# GENERATION-AUGMENTED MULTIMODAL PERSONALIZED RETRIEVAL

## ABSTRACT

With the increasing use of smartphones, users often take photos to quickly capture and store information. This multimodal personalized data offers a promising research direction for developing smartphone AI assistants. In this paper, we introduce a new task called multimodal personalized retrieval (MPR) within this context. The MPR task takes a user's text query as input and retrieves images that match the user's search intent. The task presents three key challenges: 1) effective management of personal data, 2) handling the quality of user queries, and 3) the need for a lightweight model architecture that can operate on personal devices. To address them, we propose GAMER, which enhances multimodal retrieval by leveraging LLM-driven query refinement and RLHF to optimize end-to-end performance. Extensive experiments demonstrate a 13.2% improvement over SOTA baselines. Moreover, GAMER has been deployed in real products, resulting in an improved user experience.

## 1 INTRODUCTION

Today, it has become common for users to quickly record information by taking photos with their smartphones, a practice that not only saves time but also ensures the completeness and accuracy of the information. For example, when shopping, users often capture product details, prices, or barcodes. These images can later be queried, such as asking an AI assistant (e.g., Apple's Siri), "What was the chair I saw at IKEA?" to retrieve the image, facilitating price comparison and further product exploration (as shown in Figure 1). Effectively managing and utilizing such personal data to provide services remains a challenging yet promising task. Advancements in large language models (LLMs) open new possibilities for developing personalized AI assistants on smartphones, enhancing their ability to manage and utilize user data effectively.

In this context, we introduce a new task called multimodal personalized retrieval (MPR). As depicted in Figure 1, users interact with AI assistants to recall images, such as product information captured during shopping or other life moments. Each image is associated with a description (referred to as memories) collected from user–assistant conversations, the user's social media posts, and/or content extracted from the image. Each image also includes spatio-temporal metadata (e.g., latitude, longitude, timestamp) recorded when the image was taken. This collected data, termed multimodal personalized data, facilitates the retrieval task by allowing users to input a textual query, retrieving the Top-$K$ images from their albums that best match their search intent. Unlike existing personalized LLM-based or multimodal applications, such as psychological dialogue (Zhong et al., 2024), question answering (Wang et al., 2024b), housekeeping (Han et al., 2024a), medical assistance (Zhang et al., 2023b), and web navigation (Hong et al., 2024; He et al., 2024), this task presents three unique challenges specific to real-world smartphone AI assistant scenarios, as outlined below.

(1) Multimodal Personalized Data Management: Unlike existing index structures such as HNSW (Malkov & Yashunin, 2018) and IVF (Jegou et al., 2010) used in vector databases, we observe that the personal data consists of multiple attributes, including spatio-temporal metadata, semantic memories, and visual images. This necessitates a novel structure to effectively manage the data for the personalized retrieval task, and optimize performance by leveraging these attributes. Moreover, we note that the data is organized per user and stored locally on personal devices to ensure privacy. (2) Query Text Quality: The challenge arises from user search habits, where users often input simple and highly personalized queries (e.g., just a few keywords) for retrieving images.

These brief queries make it difficult to align with potential images, leading to reduced retrieval performance. In the example shown in Figure 1, the user asks the AI assistant to retrieve "a photo taken at the airport of my son last time". This personalized query lacks clear temporal references (e.g., does "last time" mean last week or yesterday?) and person references (e.g., understanding which images contain "my son"). As a result, the assistant struggles to retrieve relevant images due to the vague nature of the query input. (3) Lightweight Model Architecture: We note that the application runs on personal devices, such as smartphones integrated with personal AI assistants. In this scenario, a lightweight model (e.g., a multimodal adapter and a smaller LLM) with fewer parameters and low FLOPs is preferred for practical deployment to end users. However, the reduction in parameters and computational resources may result in an accuracy trade-off, making a well-designed architecture essential for building the application effectively.

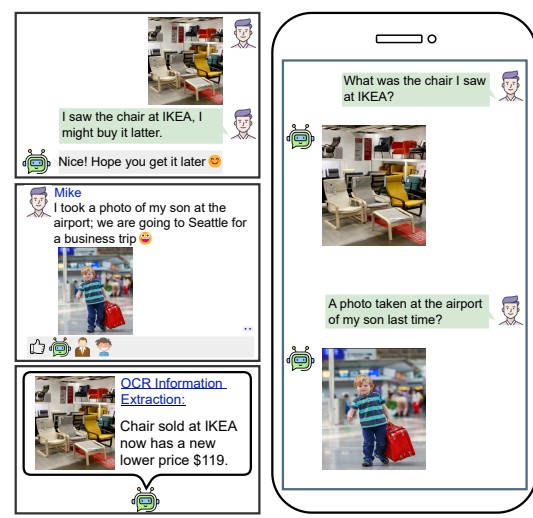

(a) Data Collection     (b) The MPR Task

Figure 1: Illustration of data collection via user-assistant conversation, social media, OCR (Left), and multimodal personalized retrieval task (Right).

To address these challenges, we introduce a new solution called **G**eneration-**A**ugmented **M**ultimodal p**E**rsonalized **R**etrieval (GAMER). Below, we discuss the solution and the rationale behind it. For (1), we propose the Multimodal Spatio-Temporal Semantic (MSTS) Index, a three-layer structure: (i) a K-D tree partitions albums into spatio-temporal cubes, (ii) a semantic graph links entities and relationships, and (iii) entity nodes store associated images and embeddings. Unlike prior multi-layer indexes (Fan et al., 2025; Chen et al., 2024c; Liu et al., 2025), MSTS is tailored for multimodal personalized data, filtering irrelevant images in early layers to enable targeted retrieval.

For (2), retrieval performance depends heavily on query quality, so we use LLMs to enrich queries with spatio-temporal cues, e.g., "last time" mapped to a temporal range for the K-D tree (Layer-1), and generating related memories to align with the semantic graph (Layer-2). A multimodal adapter further aligns vision and language, producing image embeddings matched to query embeddings (Layer-3). To personalize retrieval, we apply RLHF (Ouyang et al., 2022), where MSTS alignment tasks serve as SFT, and a reward network scores refined queries using query–image similarity and retrieval performance.

For (3), we implement a lightweight multimodal model to achieve generation objectives. Using a CLIP-VIT-Large (Radford et al., 2021) for image encoding and a smaller LLM (Radford et al., 2019; Chung et al., 2024; Yang et al., 2024a;b; Team et al., 2025) with 0.5B to 3B parameters. A Dynamic Projection Experts (DPE) adapter maps visual features into the LLM with a variable number of experts, adapting to input complexity. This minimizes overhead on personal devices by allocating resources efficiently, i.e., using fewer experts for simple inputs and more for complex ones.

To summarize, our contributions are as follows: (1) We introduce a novel task, called multimodal personalized retrieval (MPR), which presents three unique challenges compared to existing personal LLM agents. To the best of our knowledge, this is the first work of its kind. (2) We propose GAMER, a novel solution that leverages the capabilities of LLMs to enhance the retrieval process, and explicitly optimizes end-to-end performance through RLHF within the framework. (3) We conduct extensive experiments on three real-world datasets, showing significant improvements over various baselines (e.g., 13.2% improvement better than state-of-the-art baselines). Additionally, GAMER has been successfully deployed in a real smartphone AI assistant, improving the user experience.

## 2 RELATED WORK

**Personal LLM Agents.** With the rapid growth of LLMs, personal agents are being developed to access users' devices for personalized assistance. Literature includes (1) text-based agents (Zhong et al.,

2024; Wang et al., 2024b; Han et al., 2024a; Zhang et al., 2023b) and (2) multimodal agents (Shaw et al., 2023; Yan et al., 2023; Yang et al., 2023; Hong et al., 2024; Cheng et al., 2024; Dong et al., 2023), with a comprehensive survey in (Li et al., 2024). For (1), text-based agents collect user text data, including past conversations, screenshots, events, and user profiles, to support tasks such as counseling, QA, housekeeping, and medical assistance. For (2), current personal agents (Shaw et al., 2023; Hong et al., 2024; Yan et al., 2023; Yang et al., 2023; He et al., 2024) interpret multiple data types (e.g., text and images) within GUIs to simulate human interactions and accomplish tasks. In this work, we develop multimodal AI assistants for smartphones and introduce multimodal personalized retrieval, a task driven by practical scenarios.

**Vision-Language Pre-training.** Our work focuses on vision-language pre-training, where a multi-modal foundation model (Radford et al., 2021; Jia et al., 2021; Singh et al., 2022; Li et al., 2023a; 2022; Zhai et al., 2023; Xu et al., 2023; Han et al., 2024b) is trained to align image and text representations. These aligned representations are then applied to various multimodal tasks, including multimodal retrieval. In this work, we design a lightweight multimodal model optimized for personal devices, trained to align the three layers of the MSTS index to improve retrieval performance.

**Query Reformulation.** Query reformulation uses generative models to improve retrieval (Mao et al., 2021; Jang et al., 2024; Zhu et al., 2024; Mo et al., 2023; Chen et al., 2021; Xing et al., 2025; Ma et al., 2023) or conversation quality (Yuan et al., 2022). For instance, GAR (Mao et al., 2021) integrates contextual elements like answers and titles to enhance open-domain QA (Kwiatkowski et al., 2019; Joshi et al., 2017). QuARI (Xing et al., 2025) introduces a Query Adaptation Network for query-specific embedding refinement, delivering strong retrieval performance with low computational cost. Ma et al. (2023) propose a Rewrite–Retrieve–Read framework that improves retrieval-augmented LLMs by rewriting input queries to better match relevant knowledge. Our GAMER further improves queries by: (1) extracting spatio-temporal cues, (2) enriching them with personal memories, and (3) aligning query–image embeddings for more accurate retrieval. We also review multi-stage query refinement methods. CUE-M (Go et al., 2024) addresses visual QA with a RAG-style pipeline that iteratively refines queries and filters relevant information. Webwalker (Wu et al., 2025) uses a multi-agent framework where an explorer navigates web pages and a critic evaluates information sufficiency. RaFe (Mao et al., 2024) proposes a ranking-feedback–driven query rewriting framework that improves RAG retrieval quality by iteratively refining the query based on ranking signals from retrieved documents. In contrast, our work adopts a single-refinement strategy that minimizes LLM usage to improve efficiency.

**Personalized Image Retrieval.** Many recent image retrieval methods focus on generic representations, such as MoE architectures (Wang et al., 2024a), entity-centric embeddings (Wang et al., 2025), or product search (Hendriksen et al., 2022). In contrast, MPR targets personalized retrieval over user-specific memory graphs, resolving personal references (e.g., "my son" in Figure 1) that require reasoning over user context. Some works (Sun et al., 2023; Au et al., 2025; Chen et al., 2024b; Murrugarra-Llerena & Kovashka, 2019; Yeh et al., 2023; Jia et al., 2020) explore personalized retrieval. Specifically, PA-CQR (Sun et al., 2023) leverages Dynamic DPO (Rafailov et al., 2023) to align query rewriting with user preferences. PGraphRAG (Au et al., 2025) constructs a bipartite graph to model user-item interactions for the user product review generation task. In their setting, a user retrieves information from other users who have interacted with the same item. Murrugarra-Llerena & Kovashka (2019) focus on captioning and gaze prediction, showing that individuals with different personalities may view and describe the same image differently. Yeh et al. (2023) aim to localize specific moments in videos using personalized queries.Jia et al. (2020) study personalized image retrieval on Adobe Stock by modeling users and images as graph nodes connected via interaction edges (e.g., clicks, purchases). Overall, these works focus on cross-user retrieval or different problem settings, which are not applicable when user data is privately managed on individual devices.

## 3 METHODOLOGY

### 3.1 TASK FORMULATION

We formulate the task of multimodal personalized retrieval (MPR) as follows. Given a user query text $Q$, MPR aims to retrieve the user's intended images, denoted as $\mathcal{I} = \langle I_1, I_2, ..., I_K \rangle$, where $K$ represents the Top-$K$ returned images. Each stored image in the user's album $\mathbb{D}$, denoted as $I_i \in \mathbb{D}$

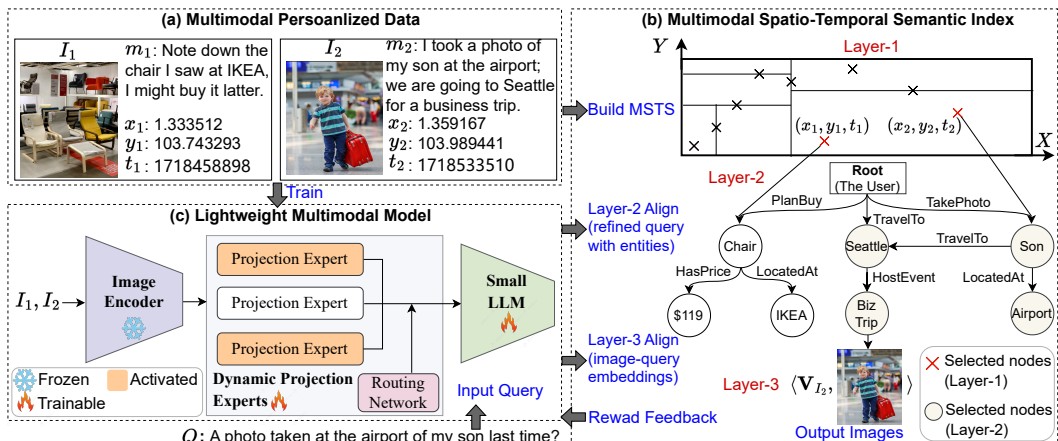

Figure 2: The architecture of GAMER: (a) Multimodal personalized data, including spatio-temporal metadata, memories, and images, supports MSTS index construction and model training. (b) The MSTS index includes a K-D tree for spatio-temporal, a semantic graph for memories, and a vector index for image embeddings. (c) The proposed multimodal model extracts spatio-temporal details, matches semantic nodes, and uses query embeddings to retrieve and rank images. Trained with RLHF, it integrates personalized generation and retrieval via reward feedback.

$(1 \leq i \leq |\mathbb{D}|)$, is associated with a tuple $(x_i, y_i, t_i, m_i)$, where $(x_i, y_i)$ represents the image's capture location (i.e., latitude and longitude), $t_i$ is the timestamp, and $m_i$ is a memory describing the image, extracted from user–assistant conversations, the user's social media posts, and/or OCR content.

## 3.2 MULTIMODAL SPATIO-TEMPORAL SEMANTIC (MSTS) INDEX

**Data Collection.** The collection of multimodal personalized data consists of: (1) images, (2) spatio-temporal metadata (captured when a user takes a photo), and (3) memories (context descriptions) related to the images. Figure 2(a) illustrates the collected personalized data in a MPR scenario, with potential data sources discussed below.

- **User–assistant conversations:** When users interact with their AI assistants, we collect the images together with the associated dialogue context.
- **Social media posts:** Users usually share album photos with captions or hashtags on social media, which naturally provide the descriptive memories. In our experiments, we use two social media datasets (Flickr and YFCC100M) as validated sources of such data.
- **OCR-based extraction:** For screenshots or photos containing rich text (e.g., product photos with price information), we apply OCR and vision–language models to extract the textual content as the memory descriptions.

**The MSTS Index Construction and Insights.** Below, we present the multilayered MSTS Index for capturing spatio-temporal, semantic, and visual attributes, with design insights in Appendix M.1 and maintenance details in Appendix M.2.

Layer-1: K-D Tree for Spatio-Temporal Metadata. For the spatial-temporal metadata $(x, y, t)$, we utilize the K-D Tree (Bentley, 1975), a space-partitioning structure for managing spatio-temporal data. Specifically, we partition the album into spatio-temporal cubes by organizing all images with a 3-dimensional K-D Tree constructed over their metadata $(x, y, t)$. Each photo is represented as a point in this space, and the K-D Tree recursively splits the album along latitude, longitude, and timestamp in a cyclic order. This recursive partitioning yields leaf nodes that correspond to small, axis-aligned regions in the spatio-temporal space. Each leaf node thus defines a spatio-temporal cube that groups images captured in nearby locations and within similar time windows.

Layer-2: Personal Semantic Graph. Next, we utilize the collected memories $m$ to construct a semantic graph that captures the user's personalized information. Entities and relationships are identified using the language processing capabilities of LLMs, and are represented as nodes and edges in the

graph, respectively. Connecting Layer-1 and Layer-2: Each stored image contains spatio-temporal information (i.e., spatio-temporal nodes) and memory information (i.e., entity nodes). We can connect spatio-temporal nodes to entity nodes if they correspond to the same image. For example, in Case 1 of Figure 2(a), we connect the spatio-temporal node $(x_1, y_1, t_1)$ to the entity nodes: Chair, 119, and IKEA. Since the 119 and IKEA nodes (child nodes) are already connected to the Chair node (parent node), it is sufficient to connect $(x_1, y_1, t_1)$ to the Chair node to traverse the entire graph. This design avoids redundant links and optimizes storage.

Layer-3: Image Vector Index. We store the images and their embedding vectors (e.g., outputs from a multimodal model discussed in Section 3.3) in this layer. Connecting Layer-2 and Layer-3: Each image contains memory information (i.e., entity nodes), and we can connect the image and its embeddings to these entity nodes. For example, in Case 1 of Figure 2(a), we connect the image $I_1$ and its embedding $\mathbf{V}_{I_1}$ to the entity nodes: Chair, 119, and IKEA. Since 119 and IKEA are leaf nodes, it suffices to link them directly to the image for retrieval. Multimodal retrieval operates at this layer by identifying images through embedding similarity search.

**Retrieval on the MSTS.** We present the retrieval process of the MPR task based on a given user query $Q$ in three steps. Step-1 (Layer-1): We identify the spatial and/or temporal context from the query and incorporate external tools (e.g., spatio-temporal NER and Geolocation API) to create a spatiotemporal cube from the identified text. This cube is then used to perform a range query (Bentley, 1975) on the K-D tree. Step-2 (Layer-2): The spatio-temporal nodes retrieved from Layer-1 activate their connected entity nodes (called activated nodes). A graph traversal (e.g., depth-first search) is then conducted starting from these activated nodes to further locate relevant entities by computing their similarities to the query. Entities with similarity scores exceeding a threshold $\delta$ are selected. Step-3 (Layer-3): We collect the images from the selected entity nodes and perform multimodal retrieval using the embeddings of the query and the images. The Top-$K$ similar images are returned and ranked based on their similarity scores.

From the above three steps, we observe that retrieval performance relies on the quality of query $Q$. This finding is consistent with intuition: the query $Q$ serves as the only input to the MPR task, and its quality directly influences the system's ability to 1) extract spatio-temporal information from $Q$ at Layer-1, 2) match relevant entities in MSTS to $Q$ at Layer-2, and 3) learn embeddings that align $Q$ with the corresponding images stored in MSTS. Building on these insights, we leverage an LLM to enhance the query content (the model architecture is introduced in Section 3.3), training it to optimize retrieval performance on the MSTS index (the training objectives are detailed in Section 3.4).

### 3.3 LIGHTWEIGHT MULTIMODAL MODEL

We propose a lightweight multimodal model architecture for personal devices, consisting of: 1) an image encoder to capture visual features, 2) a dynamic mixture of projection experts to align these visual features with query language, and 3) a smaller LLM for generating refined queries.

**Image Encoder.** We utilize the image encoder from a pre-trained vision-language model, leveraging its extensive image-text training for strong vision-language alignment. Specifically, we use CLIP-ViT-Large (428M) (Dosovitskiy, 2020; Radford et al., 2021) to extract visual features, keeping its parameters frozen during training.

**Dynamic Projection Experts (DPE).** Inspired by MoE-based LLMs (Han et al., 2024b; Huang et al., 2024), we explore a multi-expert structure to integrate visual features into the language model, leveraging its scalability and effectiveness. As illustrated in Figure 2(c), this structure comprises $M$ projection experts, each instantiated with a stack of pre-trained transformer layers. Additionally, it includes a routing network, implemented as a multilayer perceptron (MLP), which regulates the contribution of each expert to the final output. To further reduce computational costs on users' personal devices, we implement a dynamic routing strategy. Specifically, we normalize the contributions from the routing network and then sort these normalized values in descending order, denoted as $\mathbf{P} = \{P_1, P_2, \ldots, P_M\}$. We then identify the minimal number of top-ranked experts whose cumulative contribution exceeds a predefined threshold $\rho$. The number of selected experts, denoted as $t$, is determined by:

$$t = \underset{k \in \{1, 2, \ldots, M\}}{argmin} \left( \sum_{i=1}^{k} P_i \geq \rho \right). \tag{1}$$

A larger $\rho$ activates more experts, enabling adaptive selection based on routing contributions $\mathbf{P}$ and avoiding redundant computation on simple inputs.

**Smaller LLMs.** We investigate five smaller LLMs within the framework: GPT-2 (1.5B) (Radford et al., 2019), Flan-T5 (3B) (Chung et al., 2024), Qwen2 (1.5B) (Yang et al., 2024a), Qwen2.5 (0.5B) (Yang et al., 2024b), and Gemma3 (1B) (Team et al., 2025). The LLMs take as input the text prompt embeddings together with visual embeddings from the DPE, which are concatenated at the beginning of the text input sequence. This integration incorporates visual information as soft prompts, allowing the LLMs to be conditioned on visual representations for language generation.

We discuss the rationale behind the model design: 1) It represents a typical architecture of a multi-modal model, comprising an image encoder, a multimodal adapter (DPE), and an LLM. 2) To limit model size, we use smaller LLMs ranging from 0.5B to 3B. 3) We further reduce computational overhead by employing a dynamic routing strategy to activate fewer experts in the DPE.

### 3.4 ALIGNMENT OF GENERATION AND RETRIEVAL

We present the training objectives of our lightweight multimodal model, designed to generate text aligned with the MSTS index. These objectives include: 1) extracting spatio-temporal information, 2) refining the query, i.e., $\hat{Q}$, to match entity nodes in the personal semantic graph, and 3) aligning $\hat{Q}$'s embedding with stored images. As our MPR task targets personalized data retrieval, we also apply RLHF to guide the refined query generation, using a reward network trained on user image-text pairs.

**Training Aligns the MSTS Index.** These objectives are incorporated into the Supervised Fine-Tuning (SFT) stage, equipping the model with the essential capabilities to align with the index.

Spatio-Temporal Information Extraction (Layer-1). We use NER tools to extract spatio-temporal content from query $Q$; if none is found, we assign "N/A". Using the identified spatio-temporal text, we integrate geolocation APIs to construct a spatio-temporal cube, which is then used to query the K-D tree at Layer-1. If no spatio-temporal content is present, the entire K-D tree is used instead.

Image-grounded Query Refinement (Layer-2). The goal of this task is to generate memory $m$ based on the corresponding image $I$ and the query $Q$. This is essential because the personal semantic graph is constructed from $m$, which can be viewed as a refined version of the query $Q$, aligning it with the entity nodes within the graph. To achieve this, we ensure that the generated text remains semantically consistent with the visual context of the grounded image, i.e.,

$$\mathcal{L}_{\text{Layer-2}} = \sum_i - \log P(\mathbf{e}_i | \mathbf{e}_{1:i-1}, I, Q) - \sum_j^M P_j' * \log(P_j'), \tag{2}$$

where the model generates each token $e_i$ in the memory sequence $m$, conditioned on the image $I$, the query $Q$, and the previously generated tokens $e_1, ..., e_{i-1}$. Additionally, we observe that training Layer-2 may cause lazy selection in the DPE module, with all experts being chosen to improve generation quality. To address this, we introduce a regularization term in $\mathcal{L}_{\text{Layer-2}}$, which encourages selecting only the minimal necessary experts by minimizing the entropy of the expert distribution $\mathbf{P}$, where $P_j' \in \mathbf{P}$ and $M$ represents the total number of experts.

Image-Query Alignment (Layer-3). We align the representations of images and queries by bringing similar pairs closer and pushing dissimilar pairs apart using a contrastive learning approach. Specifically, we sample a batch of image-query pairs from the dataset, where each pair is denoted as $\langle I, Q \rangle$ ($I \in \mathcal{I}$ and $Q \in \mathcal{Q}$). The representations for the image and query, $\mathbf{v}_I$ and $\mathbf{v}_Q$, are derived from the output of the DPE module and the `[EOS]` token in the LLM (Radford et al., 2021), respectively. We treat $\mathbf{v}_Q$ as the positive example for $\mathbf{v}_I$ (the anchor), since they are paired, while other queries in the batch serve as negatives. The contrastive loss $\mathcal{L}_{I,Q}$ encourages alignment between the query and the anchor image by comparing the positive and negative pairs, such that:

$$\mathcal{L}_{I,Q} = \sum_{I \in \mathcal{I}} - \log \frac{\exp(\mathbf{v}_I \cdot \mathbf{v}_Q / \tau)}{\sum_{Q' \in \mathcal{Q}, Q' \neq Q} \exp(\mathbf{v}_I \cdot \mathbf{v}_{Q'} / \tau)}, \tag{3}$$

where $\tau$ represents a temperature parameter. Similarly, we can define $\mathcal{L}_{Q,I}$ by anchoring at $\mathbf{v}_Q$. The total loss $\mathcal{L}_{\text{Layer-3}}$ is then formulated as:

$$\mathcal{L}_{\text{Layer-3}} = (\mathcal{L}_{I,Q} + \mathcal{L}_{Q,I})/2. \tag{4}$$

We note that $\mathbf{v}_Q$ represents the refined query context, with similar images retrieved based on cosine similarities between these representations. The images and their trained representations are then stored in Layer-3 of the MSTS index.

**Fine-tuning for Personalization with RLHF.** We employ the PPO algorithm (Schulman et al., 2017) to fine-tune the SFT model via RLHF in order to capture personalization. The environment is modeled as a bandit setup, where a query $Q$ and its corresponding image $I$ are provided by a user, and the system generates a refined query $\hat{Q}$. The environment receives quality feedback from a reward network, and the episode concludes. The RLHF loss $\mathcal{L}_{\text{RLHF}}$ consists of two components: (1) reward and (2) KL divergence penalty. For (1), we implement a reward network based on BLIP-2 (Li et al., 2023a), which is fine-tuned on each user's image-memory pairs $\langle I, m \rangle$. The rationale is to utilize real user data to train the reward network, which assigns a score (reward $r_1$) reflecting the quality of the refined query $\hat{Q}$ by comparing it to the image $I$. This trained reward network serves as a proxy for human feedback in evaluating query reformulations; its suitability is discussed in Appendix M.4. Additionally, we consider the end-to-end performance (e.g., measured by a Recall score) of conducting the retrieval on the MSTS index, denoted as reward $r_2$. The total reward $r$ is:

$$r = \alpha * r_1 + (1 - \alpha) * r_2, \tag{5}$$

where $\alpha$ ($0 < \alpha < 1$) is a parameter that balances the effect of the two factors in the reward function. For (2), we introduce a KL divergence penalty (Jaques et al., 2019) to prevent over-optimization during the RLHF process. The RL policy $\pi_\phi^{\text{RL}}(\hat{Q}|I, Q)$ with parameters $\phi$ is penalized against the SFT model $\pi^{\text{SFT}}(\hat{Q}|I, Q)$ to maintain alignment with pre-trained behavior. The RLHF loss $\mathcal{L}_{\text{RLHF}}$ is:

$$\mathcal{L}_{\text{RLHF}} = -r + \beta \log(\pi_\phi^{\text{RL}}(\hat{Q}|I, Q)/\pi^{\text{SFT}}(\hat{Q}|I, Q)), \tag{6}$$

where $\beta$ is a coefficient to balance the strength of the reward and the KL penalty.

**Inference Stage of GAMER.** At inference, based on a user's input query, we follow the retrieval process on the MSTS index to extract spatio-temporal information, generate a refined query, and obtain its representations via the multimodal model. These outputs from the model are then utilized across the three layers of the index to retrieve the Top-$K$ images, ranked according to their similarity scores.

## 4 EXPERIMENTS

### 4.1 EXPERIMENTAL SETUP

**Dataset and Ground Truth.** We conduct experiments on three datasets: Flickr (McAuley & Leskovec, 2012), YFCC100M (Thomee et al., 2016), and Album. For Flickr, the dataset includes 268,587 images from 58,522 users. When users post image collections on the Flickr social platform, the platform records spatio-temporal metadata, short titles (used as queries), and descriptions (used as memories). We emphasize that our work uses the Flickr [1] not the image-caption dataset Flickr30k. The Flickr we used is significantly larger than Flickr30k and includes user identities, which are essential for this study. YFCC100M [2] is a publicly available multimedia dataset containing metadata for approximately 99.2 million photos. It provides user identities, timestamps, GPS information, short titles, and descriptions, making it suitable for the experimental requirements of our framework. To construct the ground truth, we note that the images in both datasets are paired with their corresponding queries, which serve as the ground truth for the MPR task. For training GAMER, 60% of each user's data is randomly selected, while the remaining 40% is reserved for testing.

In addition, we construct a new dataset (called Album) contributed by 200 data donors. Each donor provides a photo album with consent, containing anywhere from a few dozen to several thousand images. Each donor manually writes 100 queries and provides the corresponding ground-truth images. To construct personal memories, we draw on the three data sources described in Section 3.2 and guide donors to annotate their images accordingly: (1) conversational content associated with images previously chatted with AI assistants, (2) descriptions of album photos shared on their social media platforms (e.g., WeChat Moments), and (3) OCR-extracted text from photos or screenshots. For each donor, we split the 100 queries into 50% for training and 50% for testing.

---

[1] https://snap.stanford.edu/data/web-flickr.html
[2] https://multimediacommons.wordpress.com/yfcc100m-core-dataset/

**Baselines.** We categorize baseline methods into six groups, with details provided in Appendix C.

1. **Multimodal Retrieval Models**: CLIP (Radford et al., 2021), ALIGN (Jia et al., 2021), FLAVA (Singh et al., 2022), BLIP (Li et al., 2022), BLIP-2 (Li et al., 2023a), SigLIP (Zhai et al., 2023), and BridgeTower (Xu et al., 2023).
2. **Text Retrieval Models**: BM25 (Robertson et al., 1995), BGE-M3 (Chen et al., 2024a), $GTE_L$ (Li et al., 2023b), and ST-Matching (Li et al., 2023b).
3. **Query Reformulation**: GAR (Mao et al., 2021).
4. **Personalized Retrieval Models**: PA-CQR (Sun et al., 2023).
5. **Lightweight On-Device Retrieval Models**: MiniRAG (Fan et al., 2025), Reminisce (Cai et al., 2025), and GraphRAG (Edge et al., 2024).
6. **Lightweight Generative VLMs**: Qwen2.5-VL (Bai et al., 2025), LLaVa (Liu et al., 2023), and MiniCPM-V 2.6 int4 (Yao et al., 2024).

Additionally, we integrate five smaller LLMs into the GAMER framework: GPT-2 (1.5B) (Radford et al., 2019), Flan-T5 (3B) (Chung et al., 2024), Qwen2 (1.5B) (Yang et al., 2024a), Qwen2.5 (0.5B) (Yang et al., 2024b), and Gemma3 (1B) (Team et al., 2025).

**Evaluation Metrics.** We evaluate GAMER and baselines using Recall and Mean Average Precision (MAP) (Schütze et al., 2008), where higher scores indicate better performance. Additionally, we report parameter sizes (in billions) and FLOPs, following (Zhang et al., 2023a), as lower values are preferable for deployment on personal devices. All reported results are statistically significant, verified by a t-test with $p < 0.05$. See Appendix D for details.

**Implementation Details.** Implementation details of GAMER and baselines can be found in Appendix E.

### 4.2 EXPERIMENTAL RESULTS

**(1) Effectiveness Evaluation (comparison with baseline methods).** We compare GAMER with baselines using five smaller LLMs. As shown in Table 1, ALIGN and $GTE_L$ achieve the best performance among multimodal and text retrieval models, respectively. GAR and PA-CQR further enhance the results. Notably, GAMER consistently outperforms the SOTA baseline PA-CQR+$GTE_L$, e.g., achieving an 13.2% improvement on Flickr and 5.6% on YFCC100M, with consistent improvements on Album. In addition, we observe that ST-Matching is efficient but performs significantly worse than GAMER. This is because certain spatio-temporal information (e.g., time, location) embedded in text queries is difficult for embedding models to capture. Besides, we find that GAMER consistently outperforms on-device models up to 14.8%, primarily due to its design for understanding user-personalized queries with lightweight models. Similar trends are observed for recent capable generative VLMs. The five LLMs perform similarly, and we adopt Qwen2 as the default model for next experiments.

**(2) Effectiveness Evaluation (transferability test on unseen users).** We evaluate GAMER on unseen users by randomly selecting 75% users for training and the rest for testing. As shown in Table 2, GAMER consistently outperforms the baseline PA-CQR+$GTE_L$ in transferability, e.g., achieving the improvements of 11.4% on Flickr and 7.3% on YFCC100M.

**(3) Efficiency Evaluation.** As shown in Tables 1 and 2, GAMER runs comparably fast, with FLOPs consistently 13.7%, 8.6%, 12.5%, 16.8%, and 12.1% lower than PA-CQR+$GTE_L$ across the five LLMs, while maintaining superior retrieval effectiveness. Although MiniRAG runs faster than GAMER, its effectiveness is clearly worse than ours. We analyze the impact wrt the number of images per user. In Figure 3, as the number of stored images increases, retrieval time grows slightly, while Recall slightly decreases due to the larger candidate pool. PA-CQR is con-

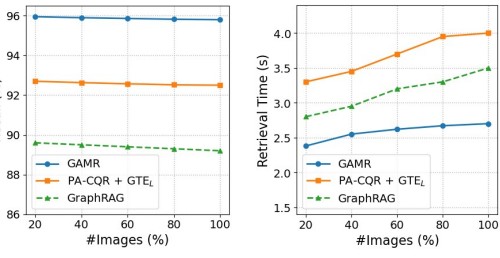

(a) Recall (#Images)     (b) Time (#Images)

Figure 3: Recall and retrieval time wrt the number of images stored per user on Flickr.

sistently lower than GAMER because, unlike GAMER, it does not align query refinement with the retrieval process along the index. Similar trends are observed on YFCC100M and Album in Appendix F.

Table 1: Comparison of GAMER (gray) with baselines; best results in bold. GAMER achieves the highest overall performance and consistently outperforms all baselines across datasets.

| Model | #Parms (B) | FLOPs (G) | Flickr Recall | Flickr MAP | YFCC100M Recall | YFCC100M MAP | Album Recall | Album MAP |
|---|---|---|---|---|---|---|---|---|
| **(1) Multimodal Retrieval Models** | | | | | | | | |
| CLIP (Radford et al., 2021) | 0.15 | 14.1 | 64.3 | 30.6 | 55.8 | 42.7 | 55.9 | 42.7 |
| ALIGN (Jia et al., 2021) | 0.17 | 10.7 | 74.7 | 53.9 | 73.1 | 60.4 | 63.1 | 50.9 |
| FLAVA (Singh et al., 2022) | 0.24 | 41.3 | 50.0 | 18.1 | 39.1 | 14.0 | 9.0 | 4.0 |
| BLIP (Li et al., 2022) | 0.22 | 64.2 | 75.0 | 49.6 | 71.6 | 50.2 | 56.8 | 45.2 |
| BLIP-2 (Li et al., 2023a) | 1.17 | 129.9 | 75.0 | 56.1 | 72.1 | 58.8 | 54.1 | 38.8 |
| SigLIP (Zhai et al., 2023) | 0.20 | 19.2 | 58.3 | 31.3 | 51.7 | 35.2 | 11.7 | 5.2 |
| BridgeTower (Xu et al., 2023) | 0.33 | 38.7 | 50.0 | 27.8 | 48.5 | 36.4 | 48.6 | 36.6 |
| **(2) Text Retrieval Models** | | | | | | | | |
| BM25 (Robertson et al., 1995) | 0.11 | 6.7 | 86.6 | 69.4 | 73.9 | 62.1 | 82.9 | 72.1 |
| BGE-M3 (Chen et al., 2024a) | 0.57 | 39.2 | 87.1 | 71.9 | 82.3 | 75.0 | 88.3 | 85.0 |
| $GTE_L$ (Li et al., 2023b) | 0.34 | 18.7 | 87.8 | 74.3 | 89.7 | 83.1 | 93.7 | 90.2 |
| ST-Matching (Li et al., 2023b) | 0.34 | 12.5 | 68.6 | 56.7 | 53.2 | 30.6 | 63.9 | 50.2 |

**(3) Query Reformulation and (4) Personalized Retrieval Models**

| Model | | #Parms (B) | FLOPs (G) | Flickr Recall | Flickr MAP | YFCC100M Recall | YFCC100M MAP | Album Recall | Album MAP |
|---|---|---|---|---|---|---|---|---|---|
| GPT-2 | GAR+ALIGN (Mao et al., 2021) | 1.78 | 15.4 | 77.3 | 55.1 | 75.5 | 65.2 | 64.1 | 52.4 |
| | GAR+$GTE_L$ (Mao et al., 2021) | 1.95 | 23.4 | 92.4 | 82.1 | 90.5 | 83.3 | 93.7 | 90.3 |
| | PA-CQR+ALIGN (Sun et al., 2023) | 1.78 | 15.4 | 76.8 | 57.5 | 75.8 | 66.3 | 68.2 | 53.4 |
| | PA-CQR+$GTE_L$ (Sun et al., 2023) | 1.95 | 23.4 | 92.4 | 87.3 | 91.0 | 84.6 | 94.2 | 90.5 |
| | GAMER | 2.13 | 20.2 | 95.4 | 90.9 | 93.7 | 89.0 | 94.5 | 91.0 |
| Flan-T5 | GAR+ALIGN (Mao et al., 2021) | 3.02 | 44.1 | 76.6 | 56.2 | 75.9 | 64.8 | 63.0 | 51.3 |
| | GAR+$GTE_L$ (Mao et al., 2021) | 3.19 | 52.1 | 91.2 | 81.4 | 90.8 | 84.7 | 93.6 | 90.5 |
| | PA-CQR+ALIGN (Sun et al., 2023) | 3.02 | 44.1 | 76.8 | 56.5 | 76.0 | 67.5 | 66.3 | 52.7 |
| | PA-CQR+$GTE_L$ (Sun et al., 2023) | 3.19 | 52.1 | 91.8 | 86.5 | 90.5 | 85.5 | 94.1 | 90.5 |
| | GAMER | 3.34 | 47.6 | 95.3 | 90.7 | 94.3 | 90.1 | 94.4 | 90.7 |
| Qwen2 | GAR+ALIGN (Mao et al., 2021) | 1.71 | 28.7 | 74.8 | 55.0 | 76.1 | 65.6 | 63.8 | 51.7 |
| | GAR+$GTE_L$ (Mao et al., 2021) | 1.88 | 36.7 | 92.0 | 77.9 | 90.9 | 84.0 | 94.0 | 90.3 |
| | PA-CQR+ALIGN (Sun et al., 2023) | 1.71 | 28.7 | 75.4 | 55.9 | 77.2 | 69.1 | 68.8 | 54.6 |
| | PA-CQR+$GTE_L$ (Sun et al., 2023) | 1.88 | 36.7 | 92.5 | 81.2 | 91.5 | 85.7 | 94.7 | 90.3 |
| | GAMER | 2.10 | 32.1 | **95.8** | **91.9** | **94.5** | **90.5** | **95.3** | **91.1** |
| Qwen2.5 | GAR+ALIGN (Mao et al., 2021) | 0.67 | 19.4 | 75.5 | 54.9 | 78.1 | 67.8 | 64.4 | 54.1 |
| | GAR+$GTE_L$ (Mao et al., 2021) | 0.84 | 27.4 | 91.9 | 80.5 | 91.0 | 83.7 | 91.2 | 89.9 |
| | PA-CQR+ALIGN (Sun et al., 2023) | 0.67 | 19.4 | 76.1 | 55.2 | 76.6 | 66.4 | 66.6 | 54.0 |
| | PA-CQR+$GTE_L$ (Sun et al., 2023) | 0.84 | 27.4 | 92.3 | 83.8 | 91.9 | 84.6 | 92.3 | 90.1 |
| | GAMER | 1.06 | 22.8 | 95.2 | 90.6 | 93.5 | 88.8 | 93.3 | 90.7 |
| Gemma3 | GAR+ALIGN (Mao et al., 2021) | 1.17 | 27.5 | 75.8 | 56.1 | 75.4 | 60.3 | 63.0 | 49.8 |
| | GAR+$GTE_L$ (Mao et al., 2021) | 1.34 | 35.5 | 90.0 | 80.4 | 90.2 | 82.2 | 93.4 | 89.2 |
| | PA-CQR+ALIGN (Sun et al., 2023) | 1.17 | 27.5 | 77.1 | 57.5 | 76.9 | 68.3 | 67.5 | 55.2 |
| | PA-CQR+$GTE_L$ (Sun et al., 2023) | 1.34 | 35.5 | 91.8 | 81.1 | 92.1 | 84.4 | 93.8 | 90.0 |
| | GAMER | 1.56 | 31.2 | 95.3 | 90.8 | 94.1 | 89.9 | 94.8 | 90.7 |

**(5) Lightweight On-Device Retrieval Models**

| Model | | #Parms (B) | FLOPs (G) | Flickr Recall | Flickr MAP | YFCC100M Recall | YFCC100M MAP | Album Recall | Album MAP |
|---|---|---|---|---|---|---|---|---|---|
| Qwen2 | MiniRAG (Fan et al., 2025) | 1.81 | 20.5 | 88.2 | 80.1 | 85.3 | 78.8 | 89.4 | 86.1 |
| | Reminisce (Cai et al., 2025) | 2.32 | 39.6 | 89.0 | 81.8 | 87.9 | 80.2 | 93.8 | 90.0 |
| | GraphRAG (Edge et al., 2024) | 2.62 | 47.9 | 89.2 | 83.3 | 88.5 | 80.8 | 94.0 | 90.1 |

**(6) Lightweight Generative VLMs**

| Model | #Parms (B) | FLOPs (G) | Flickr Recall | Flickr MAP | YFCC100M Recall | YFCC100M MAP | Album Recall | Album MAP |
|---|---|---|---|---|---|---|---|---|
| Qwen2.5-VL (Bai et al., 2025) | 3.75 | 60.5 | 89.5 | 80.8 | 88.0 | 80.3 | 93.1 | 89.6 |
| LLaVa (Liu et al., 2023) | 8.03 | 172.0 | 89.9 | 83.1 | 89.4 | 82.2 | 93.4 | 89.9 |
| MiniCPM-V 2.6 int4 (Yao et al., 2024) | 4.76 | 54.8 | 88.6 | 79.4 | 86.9 | 78.8 | 93.0 | 89.6 |

**(4) Latency across Retrieval Layers.** Balancing efficiency and effectiveness is essential for real-world deployment. Our multi-layer retrieval framework employs a lightweight top-layer filter (e.g., K-D tree) to quickly narrow down candidates, followed by more fine-grained retrieval in lower layers (e.g., personal semantic graph). This hierarchical design enables early pruning to reduce computation while maintaining accuracy. Table 3 reports the runtime of each layer, showing that the main bottleneck is Layer-2, where a smaller LLM is invoked to refine queries. Overall, the end-to-end latency (∼3s) is acceptable for personal use in practice.

Table 2: Transferability of GAMER on unseen users. GAMER maintains strong performance on unseen users, indicating good transferability of its personalized retrieval strategy.

| Model | | #Parms (B) | FLOPs (G) | Flickr Recall | Flickr MAP | YFCC100M Recall | YFCC100M MAP | Album Recall | Album MAP |
|---|---|---|---|---|---|---|---|---|---|
| **(1) Multimodal Retrieval Models** | | | | | | | | | |
| ALIGN (Jia et al., 2021) | | 0.17 | 10.7 | 66.7 | 47.5 | 63.3 | 55.6 | 62.4 | 48.1 |
| **(2) Text Retrieval Models** | | | | | | | | | |
| GTE$_L$ (Li et al., 2023b) | | 0.34 | 18.7 | 83.2 | 73.0 | 81.1 | 76.4 | 87.9 | 85.1 |
| **(3) Query Reformulation and (4) Personalized Retrieval Models** | | | | | | | | | |
| | GAR+ALIGN (Mao et al., 2021) | 1.71 | 28.7 | 72.2 | 52.8 | 71.9 | 61.0 | 62.9 | 49.9 |
| | GAR+GTE$_L$ (Mao et al., 2021) | 1.88 | 36.7 | 90.1 | 74.8 | 86.0 | 82.2 | 90.9 | 87.7 |
| Qwen2 | PA-CQR+ALIGN (Sun et al., 2023) | 1.71 | 28.7 | 73.4 | 53.2 | 72.5 | 63.4 | 64.8 | 52.1 |
| | PA-CQR+GTE$_L$ (Sun et al., 2023) | 1.88 | 36.7 | 91.3 | 79.8 | 87.9 | 83.5 | 92.9 | 89.1 |
| | GAMER | 2.10 | 32.1 | **94.4** | **88.9** | **93.3** | **89.6** | **93.8** | **89.9** |
| **(5) Lightweight On-Device Retrieval Models** | | | | | | | | | |
| | MiniRAG (Fan et al., 2025) | 1.81 | 20.5 | 84.4 | 76.4 | 81.1 | 74.9 | 83.2 | 80.1 |
| Qwen2 | Reminisce (Cai et al., 2025) | 2.32 | 39.6 | 85.5 | 80.2 | 84.4 | 76.5 | 88.6 | 85.0 |
| | GraphRAG (Edge et al., 2024) | 2.62 | 47.9 | 84.3 | 80.5 | 84.1 | 76.9 | 87.1 | 85.3 |
| **(6) Lightweight Generative VLMs** | | | | | | | | | |
| Qwen2.5-VL (Bai et al., 2025) | | 3.75 | 60.5 | 88.6 | 79.1 | 86.8 | 78.2 | 88.2 | 78.8 |
| LLaVa (Liu et al., 2023) | | 8.03 | 172.0 | 89.1 | 81.9 | 89.2 | 81.9 | 89.9 | 87.1 |
| MiniCPM-V 2.6 int4 (Yao et al., 2024) | | 4.76 | 54.8 | 87.2 | 78.3 | 86.7 | 79.0 | 88.5 | 85.3 |

Table 3: Latency across retrieval layers.

| Retrieval Layer | | Flickr | YFCC100M | Album |
|---|---|---|---|---|
| Layer-1 | Get spatio-temporal cube | 0.744s | 0.785s | 0.651s |
| | K-D tree query | 0.002s | 0.002s | 0.002s |
| Layer-2 | Query reformulation | 1.149s | 1.586s | 1.266s |
| | Graph-based retrieval | 0.470s | 0.786s | 0.353s |
| Layer-3 | Multimodal vector search | 0.352s | 0.521s | 0.303s |
| | Overall | 2.717s | 3.680s | 2.352s |

Table 4: Ablation study (Flickr).

| Components | Recall | MAP |
|---|---|---|
| GAMER | **95.8** | **91.9** |
| w/o Layer-1 | 94.1 | 90.8 |
| w/o Layer-2 | 84.5 | 81.5 |
| w/o Layer-3 | 94.9 | 90.7 |
| w/o $\mathcal{L}_{\text{Layer-2}}$ | 89.1 | 87.5 |
| w/o $\mathcal{L}_{\text{Layer-3}}$ | 94.5 | 90.8 |
| w/o $\mathcal{L}_{\text{RLHF}}$ | 94.3 | 90.6 |

**(5) Ablation Study.** Table 4 shows the impact of each MSTS layer and loss-level analysis within GAMER. Layer-2 contributes most, boosting performance by 13.4% by effectively capturing query semantics for accurate image matching. For Layer-1, the Flickr dataset provides limited explicit spatio-temporal cues due to its brief image descriptions. Accordingly, Layer-1 contributes a modest degree of search-space narrowing on this dataset. For Layer-3, the threshold $\delta$ controls the balance between Layer-2 and Layer-3. Without Layer-3, retrieval depends solely on ranking the Layer-2 candidates. With Layer-3 enabled, candidates are first filtered by $\delta$ before final ranking. We empirically set $\delta = 0.8$, which moderately constrains Layer-3's influence but yields the best overall performance. For loss-level analysis, the Layer-1 is not trained, as spatio-temporal information is obtained directly via API tools rather than learned. Omitting $\mathcal{L}_{\text{Layer-2}}$ reduces performance by about 7.5%, as the model no longer learns to refine queries. Nevertheless, the system still performs reasonably well by capturing key entities, compared with results without Layer-2. In addition, incorporating RLHF training further enhances the framework by refining query quality with personalization. This improves the accuracy of entity node matching by 1.6%, leading to more precise image retrieval.

**(6) Additional Results.** We provide additional results, including parameter studies in Appendix G, H, I, qualitative cases in Appendix J, image reduction across layers in Appendix K, and GAMER's continuous learning in Appendix L.

## 5 CONCLUSION

In this paper, we introduce a novel task of multimodal personalized retrieval aimed at developing smartphone AI assistants. Our proposed solution, GAMR, leverages LLMs through RLHF to improve retrieval performance. Experimental results demonstrate improvements of the GAMR over existing baseline methods, demonstrating real-world potential.

ETHICS STATEMENT

We provide an ethics discussion on extracting users' personal context for this new multimodal personalized retrieval application, and a broader consideration of its positive and negative impacts.

**Extracting Personal Context with LLMs.** In real-world deployment, the application leverages LLMs (e.g., GPT-4) to extract personal context from raw multimodal data. To protect user privacy, data processing must remove user identification and private information during collection, ensuring that no sensitive details are sent to external models. Furthermore, each user's data is independently collected and managed on their personal devices for enhanced protection.

In practice, we introduce two additional labels to indicate (i) the user's willingness to share information with AI assistants and (ii) the sensitivity of the private information. For data falling under low-sharing and high-sensitivity classes (e.g., important identification information, personal contact details), the information is stored and processed locally on the user's device and is not uploaded to the server for processing. In contrast, for data under high-sharing and low-sensitivity classes (e.g., schedule arrangements), server-side processing with third-party LLMs is allowed, but only with explicit user consent. Overall, the application should comply with established industry standards for data collection and processing to safeguard user privacy.

**Broader Impact.** Our work presents both potential benefits and risks. On the positive side, it enables more personalized search experiences by leveraging structured memory and multimodal context, thereby improving the intelligence and usability of smartphone applications and personal assistants. On the negative side, the reliance on personal data introduces concerns, particularly if users are unaware of how extensively their data is being utilized. This highlights the need for data transparency and user control. We believe future work should place emphasis on transparency and user awareness to mitigate potential discomfort or loss of trust.

REPRODUCIBILITY STATEMENT

Flickr and YFCC100M datasets are publicly available. We provide detailed descriptions of the GAMR architecture, hyperparameters, training setup, and baseline implementation with open-sourced code in Appendix E. Evaluation metrics, experimental configurations, and statistical analyses are reported in Section 4, ensuring reproducibility of the results. The implementation code for GAMR is currently undergoing internal review for business considerations, with plans for public release in the future.

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

## A  THE USE OF LLMs

LLMs were only used as general-purpose tools for typo correction and grammar polishing, without contributing to research ideation or experimentation.

## B    SIMILARITY AS A PROXY FOR RLHF

In standard RLHF, multiple language models (LMs) generate outputs for the same prompt, and human annotators provide pairwise preferences (e.g., "output-1 is better than output-2"). These preferences are aggregated via an ELO system (Wikipedia contributors, 2025) to train a reward model for evaluating future LM generations.

For training personalized LMs as our study, collecting user feedback for this annotation is impractical. We instead use naturally paired image-text data as a proxy, training a multimodal model to measure similarity between input images and generated queries as the reward signal.

To evaluate the alignment between our proxy reward signal (based on image-query similarity) and human preferences, we conduct a human study with 20 volunteers, following a pairwise ranking protocol inspired by standard RLHF procedures. Specifically:

1. For each participant, we randomly sample 50 sets of reformulated queries from the Flickr dataset, with each set containing 5 different reformulations of the same input.

2. Participants were asked to rank the 5 reformulated queries in each set from best to worst, based on their relevance and appropriateness with respect to the associated image and context. To simplify the process, we generated pairwise query comparisons for each participant and asked them to select the better one in each pair.

3. Based on the collected pairwise preferences, we reconstruct full rankings (e.g., $5 > 4 > 3 > 2 > 1$) using the ELO rating system (Wikipedia contributors, 2025), following the standard RLHF approach, and compare these human-derived rankings against those produced by our proposed proxy method.

4. Participants receive brief instructions on how to select their preferred reformulated query in each pairwise comparison, along with a few illustrative examples before beginning the task. No domain expertise is required.

This study involves 20 volunteers, each independently annotating 50 sets, with 5 reformulations per set, resulting in a total of 1,000 human-ranked sequences (Seq-1). For comparison, we compute system-generated rankings using our proxy reward model, which scores each reformulated query based on BLIP-2 similarity between the query and the associated image (Seq-2). To assess the alignment between human and model preferences, we calculate Kendall's tau and Spearman's rho for Seq-1 and Seq-2. The results, normalized to $[0, 1]$, are reported in Table 5. We observe a high correlation between Seq-1 and Seq-2, demonstrating that our proxy effectively approximates RLHF feedback and enables scalable personalized LM training with minimal human effort.

Table 5: Correlation between human-annotated and proxy-generated rankings.

|  | Kendall's tau | Spearman's rho |
|---|---|---|
| Correlation | $0.74 \pm 0.03$ | $0.83 \pm 0.02$ |

## C    BASELINE METHOD DETAILS

We categorize the baseline methods into six groups for comparison with the proposed GAMER.

**(1) Multimodal Retrieval Models**: These methods leverage multimodal models to generate representations for both queries and stored images. Similar images are retrieved based on the similarity between their representations. The models in this category include:

• **CLIP** (Radford et al., 2021) learns vision-language representations through contrastive learning on large-scale image-text pairs.

• **ALIGN** (Jia et al., 2021) learns joint visual-language representations from noisy image alt-text data, enabling zero-shot classification and cross-modal search without fine-tuning.

• **FLAVA** (Singh et al., 2022) targets multi-domain joint pre-training, where the foundation model learns representations from both multimodal data (image-text pairs) and unimodal data (unpaired images and text).

- **BLIP** (Li et al., 2022) adopts a multimodal encoder-decoder architecture trained with three pre-training objectives: Image-Text Contrastive Loss, Image-Text Matching Loss, and Language Modeling Loss.

- **BLIP-2** (Li et al., 2023a) introduces an efficient pre-training strategy that leverages frozen image encoders and language models, bridging the modality gap with a lightweight Querying Transformer trained in two stages.

- **SigLIP** (Zhai et al., 2023) replaces the contrastive loss used in CLIP with a pairwise sigmoid loss for aligning image-text pairs.

- **BridgeTower** (Xu et al., 2023) introduces multiple bridge layers that connect each unimodal encoder (e.g., image or text encoder) with the cross-modal encoder, aiming to enhance interaction at each layer of the cross-modal encoder.

**(2) Text Retrieval Models**: These approaches generate representations for queries and memories based on text models, where memories are linked to stored images, which are then retrieved. The models in this category include:

- **BM25** (Robertson et al., 1995) is a probabilistic information retrieval model that ranks documents based on the term frequency and inverse document frequency, optimized for relevance in search results.

- **BGE-M3** (Chen et al., 2024a) is a versatile model that supports multi-lingual, multi-functional, and multi-granularity retrieval.

- **GTE$_L$** (Li et al., 2023b) denotes a large version of the GTE models, which adopts multi-stage contrastive learning to train a unified text embedding model, achieving high efficiency and superior performance in NLP and code-related tasks.

- **ST-Matching** (Li et al., 2023b) is an intuitive method, which matches the textual Spatio-Temporal metadata (e.g., time, location) of smartphone photos with text queries. Both the textual metadata and query embeddings are obtained using GTE$_L$ (the best text model evaluated in our experiments). The Top-$K$ matching images are then retrieved based on the similarity of their embeddings.

**(3) Query Reformulation**: We consider the query reformulation model **GAR** (Mao et al., 2021), which is adapted to generate memories from input queries by fine-tuning the five smaller LLMs. The generated context is then appended to form the refined query.

**(4) Personalized Retrieval Models**: **PA-CQR** (Sun et al., 2023) is a framework that improves queries for conversational agents by learning from real user preference feedback. It uses Dynamic Direct Preference Optimization (Rafailov et al., 2023) to fine-tune language models based on which rewrites users prefer in context. In our adaptation, we fine-tune the models on our datasets and leverage our trained reward network as a proxy for user feedback to evaluate the quality of refined queries.

**(5) Lightweight On-Device Retrieval Models**: We compare GAMER with recent lightweight on-device retrieval frameworks to contextualize its trade-offs in model size, latency, and overall on-device feasibility.

- **MiniRAG** (Fan et al., 2025) is a semantic-aware, heterogeneous graph-based method that was not originally designed for multimodal retrieval. To adapt it, we extend its heterogeneous graph index by linking each user-personalized image to its corresponding entity node. The chunk segmentation nodes in MiniRAG are constructed using our personalized memory data. We then adopt MiniRAG's lightweight, graph-based knowledge retrieval process to retrieve and rank relevant images in response to user queries.

- **Reminisce** (Cai et al., 2025) is an efficient on-device multimodal embedding method. We follow the retrieval process in Reminisce by embedding images and associated metadata using a lightweight, on-device encoder to generate coarse embeddings. When a user issues a text query, it performs a fast candidate search using these coarse embeddings, then applies deeper model layers in Reminisce to refine similarity and return the most relevant images.

- **GraphRAG** (Edge et al., 2024) is adapted to the multimodal setting by linking images to the textual memories used in constructing its retrieval graph, similar to our adaptation of MiniRAG.

**(6) Lightweight Generative VLMs**: Generative VLMs, such as Qwen2.5-VL and LLaVA, are pretrained to align visual inputs with LLMs. In our approach, we fine-tune these models on the experimental training data, extract both image and user query embeddings, and perform cross-modal retrieval to identify relevant images with the query embeddings.

- **Qwen2.5-VL** (Bai et al., 2025) is a generative vision-language model that supports multi-modal understanding and generation with large-scale pretraining on both text and image data.
- **LLaVa** (Liu et al., 2023) (Large Language and Vision Assistant) is a multimodal model that aligns visual inputs with large language models to enable interactive vision-language reasoning.
- **MiniCPM-V 2.6 int4** (Yao et al., 2024) is an int4-quantized version of MiniCPM-V 2.6, optimized for single-image, multi-image, and video understanding while achieving GPT-4V-level performance on mobile devices.

## D EVALUATION METRIC DETAILS

Recall measures the fraction of relevant images retrieved within the Top-$K$ results out of all relevant images for a given query. MAP is a rank-sensitive metric that calculates the average precision (AP) for each query. AP is computed as $\sum_{k=1}^{K} P(k)\delta(k) / \sum_{i=1}^{K} \delta(i)$, where $\delta(k) = 1$ if the $k$-th image is relevant, and $\delta(k) = 0$ otherwise. $P(k)$ represents precision, which is the fraction of relevant images within the Top-$k$ results. MAP is then defined as the mean of the AP scores across all queries.

## E IMPLEMENTATION DETAILS

For the MSTS index, the parameters $\delta$ used to filter relevant entities with $\text{GTE}_L$ (Li et al., 2023b) for calculating similarities, and Top-$K$ retrieved images, are set to 0.8 and 5, respectively. We employ a caching mechanism for data preparation, where LLM inputs and outputs are cached during triplet extraction to avoid redundant LLM calls in case of interruptions or failures.

For the lightweight multimodal model, the image encoder is implemented using CLIP-ViT-Large [3]. The DPE module contains 3 experts, with each expert implemented using 8 Transformer blocks comprising 88M parameters. We set the expert allocation threshold $\rho$ to 0.7, the reward weighting parameter $\alpha$ to 0.5, and the temperature parameter $\tau$ to 0.07. The smaller LLMs, including GPT-2 [4], Flan-T5 [5], Qwen2 [6], Qwen2.5 [7], and Gemma3 [8] can be downloaded from the provided links. The overall training time is approximately 5.85 hours on Flickr, 8.21 hours on YFCC100M, and 1.66 hours on Album.We train GAMER for 20 epochs on the three datasets, with each epoch comprising 1.1K steps on Flickr, 1.6K steps on YFCC100M, and 0.4K steps on Album using an effective batch size of 128 and a maximum learning rate of 2e-5. We use the spatio-temporal NER [9] and Geolocation API[10] via the provided links.

For baseline methods, we note that all models were trained and evaluated under matched conditions, i.e., we used the same dataset splits, memory budgets, batch sizes, number of training epochs, and hardware across all compared methods, including our proposed approach. Specifically, for ALIGN [11] and $\text{GTE}_L$ [12] (the two show superior performance over the multimodal and text models, respectively), we use HuggingFace pre-trained models and follow their standard fine-tuning on our datasets. For GAR [13], we follow the official GitHub setup and use GPT-2, Flan-T5, Qwen2, Qwen2.5, and Gemma3 as backbones, matching those in GAMER for fairness. While GAR defaults to BM25, Table 1 shows

---

[3]https://huggingface.co/openai/clip-vit-large-patch14

[4]https://huggingface.co/openai-community/gpt2-xl

[5]https://huggingface.co/google/flan-t5-xl

[6]https://huggingface.co/Qwen/Qwen2-1.5B

[7]https://huggingface.co/Qwen/Qwen2.5-0.5B

[8]https://huggingface.co/google/gemma-3-1b-it

[9]https://huggingface.co/flair/ner-english-ontonotes-fast

[10]https://pypi.org/project/geopy

[11]https://huggingface.co/docs/transformers/v4.44.2/en/model_doc/align

[12]https://huggingface.co/thenlper/gte-large

[13]https://github.com/morningmoni/GAR

that ALIGN and $GTE_L$ outperform it, so we replace BM25 with these stronger retrievers to create a more competitive baseline. For PA-CQR, the detailed implementation is presented in Appendix C. For lightweight retrieval models (MiniRAG [14], Reminisce [15], and GraphRAG [16]) and generative vision-language models (Qwen2.5-VL [17], LLaVa [18], and MiniCPM-V 2.6 int4 [19]), they are available via the provided links for download and fine-tuning.

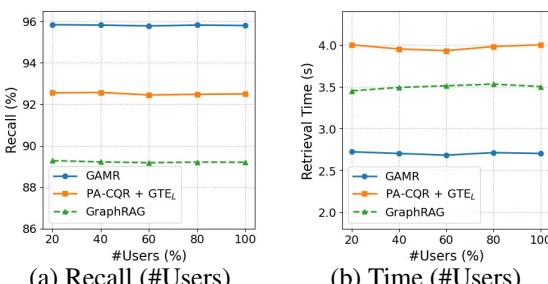

(a) Recall (#Users)  (b) Time (#Users)

Figure 4: Recall and retrieval time wrt the number of users stored per user on Flickr.

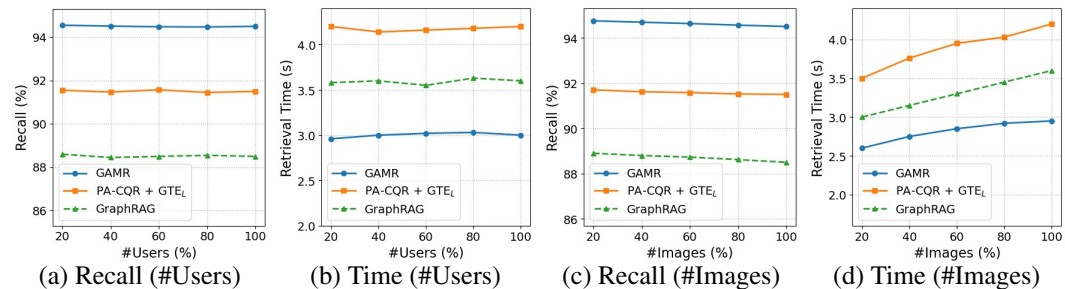

(a) Recall (#Users)  (b) Time (#Users)  (c) Recall (#Images)  (d) Time (#Images)

Figure 5: Recall and retrieval time wrt the number of users or images stored per user on YFCC100M.

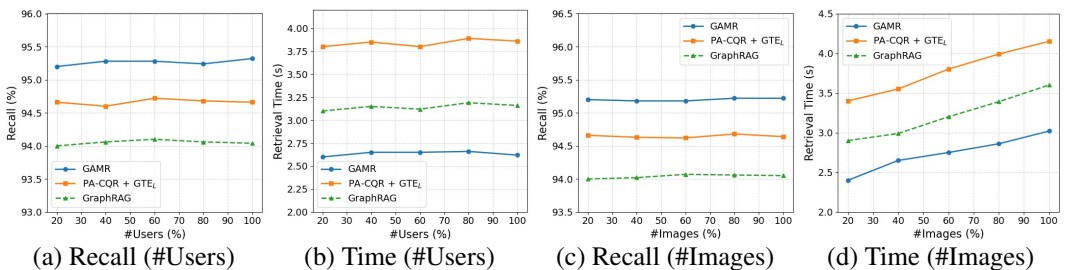

(a) Recall (#Users)  (b) Time (#Users)  (c) Recall (#Images)  (d) Time (#Images)

Figure 6: Recall and retrieval time wrt the number of users or images stored per user on Album.

# F    RETRIEVAL EFFICIENCY EVALUATION

Figure 4 presents Recall and retrieval time wrt the number of users on Flickr, and Figure 5 and Figure 6 show consistent results on YFCC100M and Album, respectively.

---

[14]https://github.com/HKUDS/MiniRAG
[15]https://github.com/caidongqi/Mobile-Search-Engine/tree/pc
[16]https://github.com/microsoft/graphrag
[17]https://github.com/QwenLM/Qwen2.5-VL
[18]https://github.com/haotian-liu/LLaVA
[19]https://huggingface.co/openbmb/MiniCPM-V-2_6-int4

Table 6: Impacts of $\delta$ for selecting entity nodes on Flickr.

| $\delta$ | 0.5 | 0.6 | 0.7 | 0.8 | 0.9 |
|---|---|---|---|---|---|
| Recall | 91.7 | 93.3 | 95.2 | **95.8** | 95.4 |
| Retrieval Time (s) | 4.3 | 3.8 | 3.2 | 2.8 | 2.6 |
| #Nodes | 10.2 | 8.9 | 7.8 | 6.5 | 5.2 |

Table 7: Impacts of $\rho$ for selecting dynamic experts on Flickr.

| $\rho$ | 0.4 | 0.5 | 0.6 | 0.7 | 0.8 |
|---|---|---|---|---|---|
| Recall | 94.9 | 95.3 | 95.5 | **95.8** | **95.8** |
| Training Time (h) | 1.39 | 1.48 | 1.56 | 1.66 | 1.75 |
| Retrieval Time (s) | 2.3 | 2.3 | 2.5 | 2.8 | 3.0 |
| FLOPs ($\times$10M) | 3207 | 3209 | 3210 | 3212 | 3214 |
| #Experts | 1.5 | 1.7 | 1.8 | 2.2 | 2.5 |

## G  PARAMETER STUDY FOR ENTITY NODE SELECTION

We vary the threshold $\delta$ from 0.5 to 0.9 and report Recall and retrieval time in Table 6. We observe that $\delta = 0.8$ achieves the best effectiveness while maintaining reasonable retrieval time. A smaller $\delta$ retains more entity nodes, which can degrade effectiveness by introducing irrelevant candidates as noise. Conversely, a larger $\delta$ filters out many relevant entity nodes, leading to reduced effectiveness. As expected, a moderate setting yields the best results.

## H  PARAMETER STUDY FOR DYNAMIC EXPERT SELECTION

We study the impact of the threshold $\rho$ in the DPE component for dynamic expert selection. A higher $\rho$ activates more experts, improving effectiveness but increasing training and retrieval costs. We set $\rho = 0.7$ to balance performance and efficiency. The projection expert maps visual features into the language space, with expert selection dynamically adjusted based on input complexity. By partitioning the visual-to-language transformation across multiple specialized experts and activating only a subset per input, our method achieves both high performance and efficiency. As shown in Table 7, selecting an average of 2.2 out of 3 experts suffices for optimal results while significantly reducing FLOPs—making the approach well-suited for deployment on personal devices.

Table 8: Impacts of $\alpha$ for reward function on Flickr.

| $\alpha$ | 0.3 | 0.4 | 0.5 | 0.6 | 0.7 |
|---|---|---|---|---|---|
| Recall | 94.2 | 94.9 | **95.8** | 94.7 | 93.8 |
| MAP | 89.7 | 91.1 | 91.9 | 90.9 | 89.5 |

## I  PARAMETER STUDY FOR REWARD FUNCTION

We vary $\alpha$ from 0.3 to 0.7 and report the performance in terms of Recall and MAP on the Flickr dataset in Table 8. We observe that when $\alpha = 0.5$, the model achieves the best results. This suggests that a balanced setup of the two factors is optimal. Therefore, we adopt this configuration as the default for reward training.

## J  QUALITATIVE RESULTS

To better understand the proposed GAMER, we present a case study in Table 9 based on the example in Figure 2. Notably, GAMER recognizes both spatial (e.g., Airport) and temporal information (e.g., a time range from the earliest timestamp in the album to the current time) from the raw query, which helps reduce the search scope. Furthermore, GAMER refines the raw query by incorporating additional entities (e.g., Seattle, business trip) learned from the user's memory. This refined query

Table 9: Case study for the example in Figure 2.

**Memory**: I took a photo of my son at the airport; we are going to Seattle for a business trip
**Raw Query**: A photo taken at the airport of my son last time?

| GAMER | PA-CQR |
|---|---|
| **Location**: Singapore Changi Airport (1.359167, 103.989441)
**Time**: [Earliest, 2024-12-15 04:25 PM (timestamp: 1718533510)]
**Refined Query**: Show the photo I took of my son at airport before our Seattle business trip | **Refined Query**: Photos of children at an airport |

more accurately aligns with its semantic graph (i.e., Layer-2 in the MSTS index), thereby facilitating the retrieval of the corresponding images.

Table 10: Candidate images reduced by each layer on Flickr.

| Top-K | 5 | 10 | 100 | 200 | 300 | 400 | 500 | 1000 | Average |
|---|---|---|---|---|---|---|---|---|---|
| # Images | 35 | 29 | 23 | 19 | 15 | 11 | 9 | 7 | 15 |
| # Images reduced by Layer-1 | 8 | 6 | 6 | 4 | 2 | 2 | 1 | 1 | 3 |
| # Images reduced by Layer-2 | 13 | 11 | 7 | 6 | 6 | 3 | 3 | 2 | 6 |
| # Remaining images in Layer-3 | 14 | 12 | 10 | 9 | 7 | 6 | 5 | 4 | 6 |

## K  IMAGE REDUCTION ACROSS RETRIEVAL LAYERS

We report the distribution of retrieved images and the reduction achieved at each layer of the GAMER pipeline, as shown in Table 10. The analysis is conducted on the Flickr dataset, where users are ordered in descending order by their image counts. For example, "Top-5" denotes the user ranked fifth in terms of total images. We also provide average statistics across all users in the dataset.

Table 11: Continuous learning for GAMER update (Flickr).

| Metric | Time-based testing | | User-based testing | |
|---|---|---|---|---|
| | Original | Fine-tuned | Original | Fine-tuned |
| Recall | 94.9 | **95.8** | 93.5 | **94.2** |
| MAP | 91.1 | **91.9** | 89.9 | **90.3** |

## L  GAMER CONTINUOUS LEARNING

We investigate updating GAMER in an online continuous learning setup on Flickr. We randomly select 500 users, each contributing 40 queries collected over the past 6 months, resulting in a total of 20,000 query-image pairs. We note that ground-truth images are available for the corresponding queries, enabling the construction of reliable query–image pairs. We conduct the experiment in two modes: 1) Time-based testing: For each user, we split their 40 queries into 30 for training and 10 for testing. The original GAMER model is evaluated on the test set. Then, we fine-tune the model using the training set and evaluate the fine-tuned GAMER on the same test set. The reported results are averaged across users. 2) User-based testing: To evaluate model transferability to unseen users, we randomly select half of the users for fine-tuning. The model is evaluated before and after fine-tuning on the remaining unseen users, and the results are again reported by averaging across users. As shown in Table 11, continuous fine-tuning yields further improvements. Similar gains are observed when updating for unseen users.

## M  FURTHER DISCUSSION

### M.1  MSTS INDEX DESIGN INSIGHTS

The design insights cover three aspects. (1) We use spatio-temporal metadata as the first layer of the index, as user queries frequently include temporal (e.g., "last week") or spatial (e.g., a specific

place name) information. This indexing layer effectively narrows down the search scope. (2) We use a graph structure to capture the relationships within a user's collected memories, where a query traverses the graph's entity nodes to retrieve semantic information for personalized retrieval, and the images corresponding to relevant entities are then gathered to facilitate multimodal retrieval. (3) We note that the multilayered design facilitates effective retrieval by filtering out irrelevant images through the consideration of spatio-temporal and semantic features, as evidenced by our ablation study results.

## M.2   MSTS Index Maintenance

The MSTS index is designed to manage users' personal data locally on their devices, where the data scale remains relatively small. MSTS combines three layers: the K-D Tree (Layer-1), a semantic graph (Layer-2, which is a type of knowledge graph), and a vector index (Layer-3). Its updates can be efficiently handled by applying standard operations and best practices for these well-established data structures (Bentley, 1975; Wang et al., 2024b; Malkov & Yashunin, 2018).

Here, we note that our MSTS index does not remove existing user photos without explicit user permission. Instead, it supports update operations that incorporate new images—such as a new chair photo—into the index. As a result, both the old and new chair images will be retrievable based on the user's query. Furthermore, if the user provides semantic feedback (e.g., indicating that the new chair photo is better than the previous one), this "better-than" relationship is updated in the graph layer of MSTS. This allows the system to prioritize the intended image in the final retrieval results, either by ranking it higher or retrieving it exclusively.

## M.3   Training under Resource Constraints on Personal Devices

We consider an edge-cloud coordination approach for training GAMER, where a subset of user-approved data is uploaded to the cloud for partial training, while privacy-sensitive data remains on the user's device for on-device training. To address resource constraints on personal devices, we consider some optimization techniques, including 1) quantized training (e.g., using 16-bit floats instead of 32-bit floats), 2) leveraging hardware acceleration, and 3) restricting training to background execution while the device is charging.

## M.4   Feasibility of RLHF for Personalization at the Individual User Level

We examine the reward mechanism in RLHF training, which leverages real user data to capture personalization. Specifically, it combines a score that measures the quality of the refined query by comparing it to the user's stored images, and incorporates the recall score to explicitly optimize personalized retrieval performance. To evaluate the effectiveness of RLHF for personalization, we conduct an ablation study (Table 4), which shows that RLHF improves overall performance, boosting recall by 1.6% on the Flickr dataset. Moreover, RLHF training effectively captures user search habits, enabling the trained GAMER model to generalize to unseen users. When tested for the transferability, it achieves over 90% recall, outperforming the SOTA baseline by 11.4% on the Flickr dataset as shown in Table 2. These results demonstrate that the personalized features learned through RLHF can generalize effectively across different users.

## M.5   How MPR Task Differs from Standard Multimodal Retrieval

Our multimodal personalized retrieval (MPR) task differs from standard multimodal retrieval in three key aspects:

- **Multilayered Retrieval**: Our approach incorporates spatio-temporal metadata (Layer-1), memory semantics (Layer-2), and image visual features (Layer-3) to enhance retrieval accuracy.
- **LLM-Driven Query Refinement**: We leverage the generative capabilities of LLMs to refine personalized textual queries, aligning them with the memory graph in Layer-2, while also training embeddings for vision-language alignment to facilitate vector search in Layer-3.
- **RLHF-Optimized Personalization**: We employ RLHF training to explicitly optimize the personalized retrieval performance through its reward mechanism.

