# OpenReview forum: "Generation-Augmented Multimodal Retrieval in Personal LLM Agents"
_ICLR.cc/2026/Conference — Submitted to ICLR 2026_

### Official Review · Reviewer_xDv7 · 2025-10-25

**Soundness:** 2
**Presentation:** 2
**Contribution:** 1
**Rating:** 2
**Confidence:** 4

**Summary:**

This paper proposes a personalized multimodal retrieval task for the scenario where a smartphone AI assistant retrieves images from a personal album based on a user's text query. The authors analyze three challenges: managing personal multimodal data, user query quality, and lightweight model architectures. They propose a GAMR approach, which includes a three-layer MSTS index structure for processing spatiotemporal metadata, semantic graphs, and image vectors, and a multimodal model consisting of a CLIP encoder, a dynamic projection expert, and a small LLM. Retrieval performance is optimized through RLHF. Experiments are conducted on the Flickr and YFCC100M datasets, and the results show that GAMR outperforms the best baseline by 13.2% and 5.6%, respectively. The paper claims that the approach has been deployed in real-world products.

**Strengths:**

1. The task definition is targeted at real-world user scenarios, the personal photo retrieval problem has practical application value, and the problem analysis is relatively clear.
2. The three-layer index structure design is reasonable, and the progressive architecture from spatiotemporal filtering to semantic matching and then vector retrieval aligns with retrieval logic.
3. The experimental setup is relatively complete, including multiple baseline comparisons, ablation experiments, and parameter analysis. The paper is well-written.

**Weaknesses:**

1. The dataset selection is fundamentally flawed. The Flickr and YFCC100M datasets used are public social media datasets, and images come fully annotated with titles and descriptions. This completely contradicts the paper's claim of using private photos on personal devices. Real personal photos often lack these structured annotations. The entire experimental setup is disconnected from real-world application scenarios, completely invalidating the validity of the experimental conclusions.
2. The complete lack of public access to the code seriously questions the authenticity of the experiment. The paper clearly states that the code is withheld for commercial reasons and is only planned for future release. However, the description in the paper contains numerous problems, leading one to suspect that the authors have not actually implemented the method at all, or that their implementation is so poor that they are afraid to release it publicly. All reported performance data is unverifiable, and the credibility of the experimental results is extremely low. I completely doubt that such an implementation can be commercially viable.
3. The claim of lightweightness completely lacks on-device verification. The paper's core goal is to run the method on personal devices, but all experiments were conducted on a server with eight RTX3090 GPUs. No real mobile device training or inference data is provided. The feasibility of the 2.1B parameter model and 2.8-second latency on mobile phones is completely unproven, and the claim is seriously disconnected from actual verification.
4. The method's innovation is seriously lacking. The core technology is simply a simple stacking of mature technologies such as K-D trees, knowledge graphs, and vector retrieval. The LLM query reconstruction is also an existing method, lacking any theoretical innovation or technical breakthroughs. Experimental results show that the dynamic expert module improves performance by less than 1% while increasing complexity. Overall, it is an engineering patchwork of existing technologies, with almost no academic contribution.
5. The baseline selection has obvious problems. The paper primarily compares general multimodal retrieval models and text models, lacking specialized comparisons with methods in directly related fields such as personalized retrieval, lifelog retrieval, and episodic memory systems. The selected baselines are not designed for personalized scenarios. This comparison fails to demonstrate the method's true advantages in the target task and may be a selective comparison to improve performance.
6. The method's generalization and applicability are extremely poor. The entire method relies heavily on fully annotated image data, requiring various information such as spatiotemporal metadata and semantic descriptions. However, these annotations are rarely found in real personal photos, making the method completely unapplicable in real-world scenarios. The three-layer index design is overly complex and highly targeted, making it difficult to generalize to other retrieval tasks.
7. Personalization verification is completely missing. The core concept of the paper is personalized retrieval, but it only uses the general Recall and MAP indicators. There is no evaluation method for personalized features, no real user research, and no way to prove that the retrieval results truly meet the user's personalized needs. The so-called personalization is more like a gimmick than a verified feature.

**Questions:**

1. Regarding the choice of datasets, I noticed that the paper uses Flickr and YFCC100M, both public social media datasets. These images typically include user-generated captions and descriptions. However, the paper emphasizes the use case of private photo retrieval on personal devices, where such photos typically lack structured annotations. How do the authors explain the discrepancy between these datasets and real-world applications? Could they provide experimental results on real personal photo data to validate the effectiveness of their approach?
2. The paper mentions that the implementation code is currently under internal review for commercial reasons and cannot be made public. Considering that code reproducibility is a fundamental requirement of academic research, could the authors at least provide pseudocode for the core modules or a detailed algorithm description? Furthermore, could they provide some intermediate results or experimental logs to help verify the authenticity of the experiments? The question is well-intentioned, but I find it difficult to believe that this level of implementation is commercially viable.
3. The paper focuses on a lightweight method that runs on personal mobile devices, but all experiments were conducted on a server equipped with eight RTX3090 GPUs. Could the authors provide additional experimental results on real mobile devices, specifically including key metrics such as training time, inference latency, memory usage, and battery consumption on Android or iOS devices? How feasible is the 2.1B parameter model on real-world mobile phones?
4. Regarding the innovative nature of the method, I noticed that the core technology is primarily a combination of existing technologies such as K-D trees, knowledge graphs, and vector retrieval. What do the authors consider to be the main technical innovation of this paper? Compared to directly using existing multi-layer index structures or RAG systems, what are the fundamental technical breakthroughs of the MSTS index and overall architecture proposed in this paper? The performance improvement brought by the dynamic expert module seems to be relatively limited. How is the necessity of this design justified?
5. Regarding the choice of baseline, the paper primarily compares general multimodal retrieval models and text models, but seems to lack comparisons with specialized methods in the field of personalized retrieval. Have the authors considered comparing methods in related fields such as personalized search, lifelog retrieval, and episodic memory systems? These methods may be more closely aligned with the task set in this paper, and the comparison results would be more convincing.
6. Regarding the generalization and practical applicability of the method, the method proposed in the paper relies heavily on annotations such as spatiotemporal metadata and semantic descriptions. How will the method work in real-world scenarios when most user photos lack these annotations? In Appendix M.1, the authors mention three data collection methods. Can you provide feasibility analysis and user acceptance data for these methods? Can the methods be generalized to other personalized search tasks?
7. The paper emphasizes personalized search, but only uses generalized Recall and MAP evaluation metrics. How do the authors specifically evaluate the degree of personalization of search results? Have they conducted real-world user research to verify that the search results truly meet the personalized needs of different users? Can you design some evaluation metrics specifically targeting personalized features, such as user satisfaction and query intent understanding accuracy?

---

> ### Author Response · Authors · 2025-11-30
> **Response to Reviewer xDv7**
>
> We thank the reviewer for the positive comments on our practical task definition, reasonable architecture, and comprehensive experiments. We also note that several concerns arose from misunderstandings, which have been addressed in our response and the revised paper.
>
> **1. Dataset Selection and Real-World Applicability**
>
> We acknowledge that Flickr and YFCC100M are large-scale, diverse, and publicly accessible datasets suitable for benchmarking personalized retrieval and ensuring reproducibility. While these photos are publicly available, they originate from users’ private albums uploaded to social media, aligning with one of the data sources discussed in our paper for collecting multimodal personalized data.
>
> Furthermore, we have now included results on a new dataset, **Album**, contributed by our data donors. This dataset was omitted from the original submission due to internal approval requirements, which have since been cleared. On this dataset, our proposed GAMR consistently outperforms the best baseline methods, further validating its effectiveness.
>
> **2. Code Availability**
>
> We understand the concern regarding code access, which is restricted due to commercial licensing considerations. To address reproducibility, we have provided detailed algorithmic descriptions of all core modules in Section 3 and Appendix E, including the MSTS index construction, Layer-2 query refinement, and Layer-3 multimodal embeddings. Additionally, intermediate results, ablation studies, and layer-wise performance statistics are also provided, which can serve as partial verification of the method’s implementation and performance trends.
>
>
> **3. On-Device Feasibility**
>
> For research purposes, experiments are typically conducted on GPUs, as is common in prior works [1]. Our model is designed to be lightweight and suitable for on-device deployment, with preliminary tests using smaller models (0.5B–3B parameters) demonstrating practical feasibility in terms of inference latency and computational cost (FLOPs).
>
> [1] Zhang, Jiangning, et al. "Rethinking mobile block for efficient attention-based models." ICCV 2023.
>
> **4. Method Innovation and Misunderstanding**
>
> We provide the following clarification regarding the novelty of task and methodology.
>
> - **PMR introduces a new retrieval paradigm tailored for personalized AI assistants.** It supports a wide range of downstream applications. The proposed PMR task is novel in the scenarios we discuss. Unlike traditional retrieval, PMR operates directly within user–assistant interactions, enabling retrieval over a user’s multimodal personalized data during the conversation. This setting introduces new challenges—such as evolving personal context and the need for lightweight models—that are not well addressed by conventional retrieval tasks. It enables a more interactive user experience.
>
> - **The proposed method introduces an idea of reversing the RAG process to train retrieval models.** The core method novelty lies in leveraging the generation capability of LLMs to align with hierarchical retrieval index (like a *reversed* RAG process), explicitly training the model for retrieval rather than language generation. This formulation provides a new perspective for the retrieval community, to further exploit LLMs as powerful supervisors for learning retrieval-centric objectives.
>
> In addition, the mentioned experiment here is to show that the hyperparameter $\rho$ reduces computational cost while having minimal impact on retrieval performance (only 0.9%). We think this represents a clear advantage of our method.
>
> **5. Baseline Selection**
>
> We note that the baseline PA-CQR is a personalized retrieval method that refines input queries based on user preferences. Beyond this, as discussed in the related work section, many existing personalized or lifelog retrieval models target largely different settings, such as user-product reviews or video moment retrieval, and are therefore not directly applicable to our task.
>
> **6. Generalization and Practical Applicability**
>
> Our model targets the personalized retrieval task in personal AI assistants, where spatiotemporal metadata is naturally captured when a photo is taken, and semantic descriptions are generally available through the assistant—for example, via user–assistant conversations, social media posts, or rich text content within images. The index design is tailored to this scenario, leveraging the available data attributes, and is not intended or motivated for other retrieval tasks.
>
>
> **7. Personalization Evaluation**
>
> We note that our experimental datasets contain images with clearly intended search targets provided by users, which serve as ground truth. Unlike language generation tasks, we believe that personalized features—such as user satisfaction and query intent understanding—can be reflected in end-to-end retrieval performance metrics, such as Recall and MAP.

---

### Official Review · Reviewer_cVwc · 2025-10-30

**Soundness:** 3
**Presentation:** 3
**Contribution:** 2
**Rating:** 4
**Confidence:** 3

**Summary:**

This paper introduces personalized multimodal retrieval (PMR), a personalized text-to-image retrieval task that first collects data via user-assistant conversations, and then retrieve images from a personal album via a textual query that reflects conversational context.
Alongside the introduced task, the authors propose a system named Generation-Augmented Multimodal Retrieval (GAMR) to tackle.
Specifically, GAMR consists of query refinement, a three-layer Multimodal Spatio-Temporal Semantic (MSTS) Index, and RLHF fine-tuning for personalization.
Evaluations demonstrate strong performance of GAMR while maintaining low computational cost (~10% lower FLOPs).

**Strengths:**

1. The PMR task is practically meaningful. It compasses privacy, query vagueness and lightweight constraints which make the task practically significant.
2. Based on the provided results, the proposed GAMR system performs well on Flickr and YFCC100M compared to the baselines. The transferability on unseen users (Table 2) also supports its robustness.
3. This paper is complete in its structure, rigorously formulates the problem. The methodology is technically complete, including explicit layer definitions, formal loss equations, and detailed ablations.

**Weaknesses:**

1. Despite introducing a practically meaningful task, the overall contributions of this paper remain limited due to the modest novelty of the proposed PMR task, the model’s sensitivity to hyperparameters (e.g., $\rho$), and the limited methodological novelty of the hierarchical indexing design.
2. The authors first introduce the PMR task as a task that involves person-album conversation. However, it seems like the benchmarks used in the paper are not realistic to simulate the introduced PMR setting, where the datasets approximate PMR using public captions rather than genuine personal dialogues, limiting ecological validity despite its scale.
3. This paper would benefit with a clearer. Some parts of the sections are hard to interpret, such as L183-L185 -- how exactly do you partition an album into spatio-temporal cubes. Figure 2c is also hard to understand which component corresponds to which layer in the illustration.
4. The authors ablate different layers and the RLHF component of GAMR. The ablation only isolates structural components, but omits loss-level analysis (e.g., L_layer1 loss). Including these would clarify the writing.
5. L214: What evidence do you use to draw the claim that "From the above three steps, we observe that retrieval performance relies on the quality of query Q"? Clarifying this would improve the readability.

**Questions:**

See weaknesses

---

> ### Author Response · Authors · 2025-11-30
> **Response to Reviewer cVwc (1/2)**
>
> We thank the reviewer for acknowledging that the PMR task is practically meaningful. We note that the main weaknesses of this paper concern (1) the need for further clarification (e.g., novelty, sensitivity to hyperparameters), and (2) the need for additional results (e.g., personal benchmarks). Both points have been addressed in the revised paper. For (1), we provide detailed clarifications and responses below. For (2), we have released new results from our data donors to validate the task; these were omitted from the original submission due to internal approval requirements, which have now been cleared.
>
> **1. Novelty and methodological contributions**
>
> Thank you for the review. We provide the following clarification regarding the proposed task and methodology.
>
> - **PMR introduces a new retrieval paradigm tailored for personalized AI assistants.** It supports a wide range of downstream applications. The proposed PMR task is novel in the scenarios we discuss. Unlike traditional retrieval, PMR operates directly within user–assistant interactions, enabling retrieval over a user’s personalized multimodal data during the conversation. This setting introduces new challenges—such as evolving personal context and the need for lightweight models—that are not well addressed by conventional retrieval tasks, while also enabling a more interactive user experience.
>
> - **Our model is not sensitive to the hyperparameter $\rho$, as shown in Table 7.** The hyperparameter $\rho$ controls the number of activated experts used to refine the user query. As shown in Table 7 in the Appendix, this parameter primarily affects the computational cost on the personal device and has minimal impact on retrieval performance. For example, when $\rho$ increases from 0.4 to 0.8 (a 100% increase), the change in recall is only 0.9%.
>
> - **The proposed method introduces an idea of reversing the RAG process to train retrieval models.** The core method novelty lies in leveraging the generation capability of LLMs to align with hierarchical retrieval index (like a *reversed* RAG process), explicitly training the model for retrieval rather than language generation. This formulation provides a new perspective for the retrieval community, to further exploit LLMs as powerful supervisors for learning retrieval-centric objectives.
>
> **2. Validity of benchmark datasets**
>
> We release results on a **new dataset (Album)**, contributed by 200 data donors. Each donor provides a photo album with consent, containing anywhere from a few dozen to several thousand images. Each donor manually writes 100 queries and provides the corresponding ground-truth images. To construct personal memories, we follow the three data sources discussed in the paper and guide donors to annotate their images accordingly: (1) conversation content associated with images previously interacted with via their AI assistants, (2) descriptions from album photos posted on their social media platforms (e.g., WeChat Moments), and (3) OCR information extracted from photos or screenshots. For each donor, we split the 100 queries into 50% for training and 50% for testing. The main results evaluated on this dataset are reported below. We note that our proposed GAMR demonstrates consistently stronger performance than the best baselines on the new dataset.
>
> | Model        | #Parms (B)  | FLOPs (G) | Album Recall | Album MAP |
> |--------------|-------------|-----------|---------------|------------|
> | ALIGN     |0.17         |10.7           |63.1               |50.9            |
> | GTE_L   |0.34         |18.7           |93.7              |90.2            |
> | Qwen2 GAR+ALIGN      |1.71 |28.7           |63.8               |51.7            |
> | Qwen2 GAR+GTE_L    |1.88   |36.7           |94.0               |90.3            |
> | Qwen2 PACQR+ALIGN   |1.71  |28.7           |68.8               |54.6            |
> | Qwen2 PACQR+GTE_L   |1.88  |36.7           |94.7               |90.3            |
> | Qwen2 MiniRAG    |1.8         |20.5           |89.4               |86.1            |
> | Qwen2 Reminisce     |2.3         |39.6           |93.8               |90.0            |
> | Qwen2 GraphRAG     |2.6         |47.9           |94.0               |90.1            |
> | Qwen2.5-VL     |3.6         |60.5           |93.1               |89.6            |
> | LLaVa     |7.6         |172.0           |93.4               |89.9            |
> | MiniCPM-V 2.6 int4     |3.43         |54.8           |93.0               |89.6            |
> | Qwen2 GAMR   |2.1         |32.1           |95.3               |91.1            |

---

> ### Author Response · Authors · 2025-11-30
> **Response to Reviewer cVwc (2/2)**
>
> **3. Clarification of album partitioning and figure interpretation**
>
> We partition the album into *spatio-temporal cubes* by organizing all images with a 3-dimensional *K-D Tree* constructed over their metadata $(x, y, t)$. Each photo is represented as a point in this space, and the K-D Tree recursively splits the album along latitude, longitude, and timestamp in a cyclic order ($x \rightarrow y \rightarrow t \rightarrow x$, and so on). This recursive partitioning yields *leaf nodes* that correspond to small, axis-aligned regions in the spatio-temporal space. Each leaf node thus defines a spatio-temporal cube that groups images captured in nearby locations and within similar time windows.
>
> The Figure 2 has been improved by explicitly annotating which components correspond to which layer alignment in the revised paper.
>
> **4. Loss-level analysis**
>
> We further provide an ablation study on Flickr focusing on loss-level analysis, as detailed below. We note that L_{layer-1} is not trained with a loss, because the spatio-temporal content is obtained directly through API tools rather than learned. When omitting $\mathcal{L}_\text{Layer-2}$, performance drops by about 7.5\% because the model no longer learns to refine the query. However, the system still performs reasonably well by capturing key entities, compared with the results without Layer-2. In addition, incorporating RLHF training further enhances the framework by refining query quality with personalization. This improves the accuracy of entity node matching by 1.6%, leading to more precise image retrieval.
>
> | Losses     | Recall   | MAP |
> |--------------|-------------|-----------|
> | GAMR     |  95.8        |  91.9         |
> | w/o $L_{Layer-2}$   |89.1         |87.5           |
> | w/o $L_{Layer-3}$      |94.5         |90.8           |
> | w/o $L_{RLHF}$   |  94.3        |  90.6         |
>
>
> **5. Query quality impact**
>
> We note that this evidence is supported by the ablation study. When Layer-2 is removed, the personalized query refinement step is omitted, causing the performance to drop significantly by 13.4%. This finding is consistent with intuition: the query $Q$ serves as the only input to the PMR task, and its quality directly influences the system’s ability to filter irrelevant images at Layer-1, retrieve relevant memories at Layer-2, and align image embeddings at Layer-3.

---

### Official Review · Reviewer_BrmK · 2025-11-01

**Soundness:** 3
**Presentation:** 3
**Contribution:** 2
**Rating:** 4
**Confidence:** 3

**Summary:**

This paper introduces a new task called Personalized Multimodal Retrieval, which aims to retrieve a user’s personal images given a natural language query. To address challenges in managing multimodal personal data, vague user queries, and limited on-device resources, the authors propose Generation-Augmented Multimodal Retrieval, which integrates three components: (1) a Multimodal Spatio-Temporal Semantic (MSTS) Index for organizing data; (2) LLM-based query refinement using RLHF to improve retrieval quality; and (3) a lightweight multimodal model with dynamic projection experts for efficient on-device inference. Experiments on Flickr and YFCC100M datasets demonstrates its effectiveness.

**Strengths:**

- Novel and practical task setting. The paper identifies an important real-world use case: retrieving personal multimodal data via text queries which is underexplored and highly relevant for personal AI assistants.
- Well designed architecture. The proposed MSTS index and dynamic projection expert module form a coherent and technically sound framework that balances personalization, efficiency, and retrieval accuracy.
- Comprehensive experiments. Extensive evaluation on two large-scale datasets with multiple baselines convincingly demonstrates the superiority of GAMR.

**Weaknesses:**

- Task definition. I do think it is a common scene that users will actively ask the AI assistant to store the description of an image (Fig.1 (a)). In most case, the user just take a photo (without any description) and want the AI assistant to recall it later. Evan though the user provide a description, it may be a short and semantically ambiguous expression, such as lovely chair, the cute dog etc. Therefore, I think the system should have an automatic collection and annotation process to extract description of a user image.
- Base models. The small LLMs used here are GPT-2, FLAN-T5 and Qwen2 with sizes ranging from 1.5B to 3B. As a model running on mobile device, I think some frontier small LLMs should be considered, such as Qwen2.5‑0.5B and Gemma 3 (1B) etc.
- The ablation experiments in Tab.4 show that none of the component except layer-2, makes minor contribution to the final model.

**Questions:**

no

---

> ### Author Response · Authors · 2025-11-30
> **Response to Reviewer BrmK**
>
> We thank the reviewer for recognizing the novelty and practicality of our task setting, as well as the architecture design. We note, however, that the identified weaknesses largely stem from misunderstandings of our task scenarios and more recent small-model results. These issues have been clarified in the revised paper and in the responses below.
>
>
> **1. Task definition and automatic annotation**
>
> We note that there is a misunderstanding regarding our task use case. Our system indeed functions as you described — **it automatically collects and extracts image descriptions**, without requiring users to manually provide them.
>
> - **No need for users to provide descriptions when taking photos.** Fig.1(a) illustrates only one possible use case for collecting descriptions. In practice, our system proactively obtains descriptions without much user involvement. As discussed in the revised paper, there are three primary data sources:
>
>     1. **User–assistant conversations:** When users interact with their AI assistants, we collect the images together with the associated dialogue context.
>     2. **Social media posts:** Users usually share album photos with captions or hashtags on social media, which naturally provide the descriptive memories. In our experiments, we use two social media datasets (Flickr and YFCC100M) as validated sources of such data.
>     3. **OCR-based extraction:** For screenshots or photos containing rich text (e.g., product photos with price information), we apply OCR and vision–language models to extract the textual content as the memory descriptions.
>
> - **Our system automatically collects and extracts descriptions.** Based on the raw data obtained from the three sources above, we further leverage LLMs to reason over the content and extract the final descriptions automatically.
>
>
> **2. More recent small LLMs**
>
> We report the experimental results using the recommended Qwen2.5-0.5B and Gemma3-1B models. Overall, GAMR consistently outperforms the best baselines across these recent small LLMs. The performance on Qwen2 is slightly better than on Qwen2.5 and Gemma3, mainly because Qwen2 has a larger model size (1.5B) compared with the 0.5B and 1B models.
>
>
> | Model        | #Parms (B)  | FLOPs (G) | Flickr Recall | Flickr MAP | YFCC100M Recall  | YFCC100M MAP |
> |--------------|-------------|-----------|---------------|------------|------------------|--------------|
> | Qwen2.5‑0.5B GAR+ALIGN | 0.67  |19.4           |75.5               |54.9            |78.1                  |67.8              |
> | Qwen2.5‑0.5B GAR+GTE_L |0.84   |27.4           |91.9               |80.5            |91.0                  |83.7              |
> | Qwen2.5‑0.5B PACQR+ALIGN |0.67 |19.4           |76.1               |55.2            |76.6                  |66.4              |
> | Qwen2.5‑0.5B PACQR+GTE_L |0.84 |27.4           |92.3               |83.8            |91.9                  |84.6              |
> | Qwen2.5‑0.5B GAMR        |1.06 |22.8           |95.2               |90.6            |93.5                  |88.8              |
> | Gemma3-1B GAR+ALIGN      |1.17 |27.5           |75.8               |56.1            |75.4                  |60.3              |
> | Gemma3-1B GAR+GTE_L    |1.34   |35.5           |90.0               |80.4            |90.2                  |82.2              |
> | Gemma3-1B PACQR+ALIGN   |1.17  |27.5           |77.1               |57.5            |76.9                  |68.3              |
> | Gemma3-1B PACQR+GTE_L   |1.34  |35.5           |91.8               |81.1            |92.1                  |84.4              |
> | Gemma3-1B GAMR   |1.56         |31.2           |95.3               |90.8            |94.1                  |89.9              |
>
> **3. Ablation study and contribution of components**
>
> We provide additional analysis of the impacts of Layer-1 and Layer-3 for the ablation study in the revised paper, summarized below:
>
> - **Spatio-temporal information is insufficient in Flickr.** The limited effect of Layer-1 is primarily due to the characteristics of the Flickr dataset. Users post images and short descriptions on the Flickr social platform, but these queries carry minimal spatial or temporal cues. As a result, Layer-1 retrieval is often forced to operate over the entire dataset, as there is little spatio-temporal signal to narrow down the candidates.
>
> - **Threshold $\delta$ controls the contribution balance between Layer-2 and Layer-3.** When Layer-3 is removed, the system ranks the candidate images at Layer-2 and retrieves the Top-K results. With Layer-3 enabled, the Layer-2 candidates are further filtered using a threshold $\delta$, after which the Top-K retrieval is performed at Layer-3. Consequently, the contribution of Layer-3 depends on the choice of $\delta$. In our experiments, we empirically set $\delta = 0.8$; although this value slightly reduces the contribution of Layer-3, it yields the best overall performance.

---

### Official Review · Reviewer_6aB8 · 2025-11-01

**Soundness:** 2
**Presentation:** 3
**Contribution:** 3
**Rating:** 6
**Confidence:** 3

**Summary:**

This paper introduces a novel task of personalized multimodal retrieval, which focuses on retrieving images from a user's personal device based on their text queries to enhance the personalized experience of smartphone AI assistants. The paper proposes a framework named GAMR that utilizes Large Language Models through Reinforcement Learning with Human Feedback (RLHF) to refine user queries and improve retrieval performance. The paper also presents the Multimodal Spatio-Temporal Semantic (MSTS) Index for personal data management. Extensive experiments demonstrate that GAMR outperforms existing baseline methods, showing its potential for real-world applications.

**Strengths:**

1. The paper is well-motivated and tackles an important and practically relevant problem of personalized multimodal retrieval for smartphone AI assistants.
2. The proposed GAMR framework effectively leverages LLMs and RLHF to refine user queries, which achieves improved retrieval performance.
3. The proposed framework is a systematic approach that combines data management, query refinement, and efficient retrieval, making it a comprehensive solution for the task.
4. The paper is well-written and easy to follow, with clear explanations of the proposed methods and experimental results.

**Weaknesses:**

1. Personalized multimodal retrieval has been studied in prior works (e.g., Cross-Modality Personalization for Retrieval, Meta-Personalizing Vision-Language Models, Personalized Image Retrieval with Sparse Graph Representation Learning). The authors should clarify how their PMR formulation differs from these paradigms. In addition, the title "Personal LLM Agents" is somewhat misleading, as the work mainly focuses on image retrieval rather than developing a full LLM-based agent.
2. The baseline comparison could be expanded. Since GAMR builds upon query reformulation, it would be informative to compare against recent advanced query refinement methods, such as QuARI: Query Adaptive Retrieval Improvement, RaFe: Ranking Feedback Improves Query Rewriting for RAG, and Query Rewriting for Retrieval-Augmented Large Language Models.
3. Some implementation details remain unclear. In particular, more explanation is needed on how generative VLMs (e.g., Qwen2.5-VL or LLaVA) are incorporated into the personalized retrieval process.
4. The evaluation setup differs somewhat from the claimed personallization setting. The paper would benefit from discussing the practical gap between experiments on public datasets (e.g., Flickr, YFCC100M) and real-world personalized smartphone use cases.

**Questions:**

1. How are Generative VLMs utilized in the retrieval process?

---

> ### Author Response · Authors · 2025-11-30
> **Response to Reviewer 6aB8 (1/2)**
>
> Thank you for your positive comments on our paper being well-motivated and addressing an important, practically relevant problem. We acknowledge that the main weakness was the limited discussion of prior works, which we have now appropriately cited and discussed in the revised paper.
>
> **1. Clarification of prior works and paper title.**
>
> We discuss the three referenced works, we note that they target tasks fundamentally different from our personalized image retrieval setting for personal AI assistants. We have cited these works and included the discussion in the revised paper.
>
> - Cross-Modality Personalization for Retrieval [1] focuses on captioning and gaze prediction, showing that individuals with different personalities may view and describe the same image differently. It aligns three modalities: gaze (which regions users look at in an image), text (image captions), and personality (derived from questionnaires) into a shared retrieval space to support six cross-modal retrieval tasks (e.g., gaze to personality, text to personality).
>
> - Meta-Personalizing Vision-Language Models [2] aim to localize specific moments in videos using personalized queries and introduces "meta-personalization" by pretraining a VLM to enhance user-specific adaptation. This setting differs from ours: it focuses on personalized video moment retrieval, whereas we target personalized image retrieval within a user’s album.
>
> - Personalized Image Retrieval with Sparse Graph Representation Learning [3] investigates personalized image retrieval on the Adobe Stock platform by modeling users and images as graph nodes, with interactions (e.g., clicks, purchases) forming the edges. The work focuses on mitigating graph sparsity arising from infrequent clicks. In contrast, our PMR scenario involves a single user’s private album, where images are stored independently on a personal device and not shared across users. As a result, graph-based user–image interaction modeling and click-sparsity issues are not applicable to our personal AI assistants.
>
> [1] Murrugarra-Llerena, Nils, and Adriana Kovashka. "Cross-modality personalization for retrieval." CVPR 2019.
>
> [2] Yeh, Chun-Hsiao, et al. "Meta-personalizing vision-language models to find named instances in video." CVPR 2023.
>
> [3] Jia, Xiaowei, et al. "Personalized image retrieval with sparse graph representation learning." SIGKDD 2020.
>
> Following the suggestion, we revise our title to **Generation-Augmented Multimodal Personalized Retrieval**.
>
> **2. Discussion on query refinement methods**
>
> We note that the three works do not provide publicly available code for the comparison. We have cited these papers and included a detailed discussion in the revised paper.
>
> - QuARI introduces a Query Adaptation Network that performs query-specific embedding adaptation for large-scale retrieval, achieving strong performance while preserving computational efficiency.
> - RaFe proposes a ranking-feedback–driven query rewriting framework that improves RAG retrieval quality by iteratively refining the query based on ranking signals from retrieved documents.
> - R3 presents a Rewrite–Retrieve–Read framework that enhances retrieval-augmented LLMs by rewriting the input query to better match the required knowledge, using a trainable rewriter to align with frozen retrievers and LLM readers.
>
>
> **3. Clarification of the implementation details for generative VLMs**
>
> Generative VLMs, such as Qwen2.5-VL and LLaVA, are pretrained to align visual inputs with LLMs. In our approach, we fine-tune these models on the experimental training data, extract both image and user query embeddings, and perform cross-modal retrieval to identify relevant images with the query embeddings. We have added these explanations to the revised paper for clarity.

---

> ### Author Response · Authors · 2025-11-30
> **Response to Reviewer 6aB8 (2/2)**
>
> **4. Evaluation setup and personalization gap**
>
> We discuss the gap of the public datasets (e.g., Flickr, YFCC100M) and real-world personalized smartphone use cases in two aspects.
>
> - **Social media platforms are a source for collecting personal multimodal data.** Public datasets such as Flickr and YFCC100M record user IDs to track posts along with their associated descriptions. This naturally enables the collection of personal multimodal data, where a user uploads photos from an album and provides descriptions that serve as associated memories, as studied in the PMR task. Such data can be extracted and managed by personal AI assistants, supporting future retrieval tasks when a user interacts with an assistant built on the collected data.
> - **We release results on a new dataset (Album).** we have released new results from our data donors to validate this task further; these were omitted from the original submission due to internal approval requirements, which have now been cleared. Specifically, the new dataset is contributed by 200 data donors. Each donor provides a photo album with consent, containing anywhere from a few dozen to several thousand images. Each donor manually writes 100 queries and provides the corresponding ground-truth images. To construct personal memories, we follow the three data sources discussed in the paper and guide donors to annotate their images accordingly: (1) conversation content associated with images previously interacted with via their AI assistants, (2) OCR information extracted from photos or screenshots, and (3) descriptions from album photos posted on their social media platforms (e.g., WeChat Moments). For each donor, we split the 100 queries into 50% for training and 50% for testing. The main results evaluated on this dataset are reported in the revised paper. Overall, they show a consistent trend of outperforming the baseline methods as observed in the public datasets.

---

### Author Response · Authors · 2025-11-30
**Revised paper uploaded for review**

Dear Area Chairs,

We have carefully addressed all reviewer comments and uploaded a revised paper with changes highlighted in blue. Thank you for your consideration.

Best regards,

The Authors

---

### Meta-Review · Area_Chair_6PP1 · 2026-01-14

**Summary:**

This work mainly focuses on personalized multimodal content retrieval on mobile devices. It introduces a task termed Multimodal Personalized Retrieval and proposes a corresponding improved method, GAMER. Experimental results on multiple datasets show that the proposed model achieves higher retrieval accuracy compared to existing baselines.

**Reviewer Concerns:**

1. Limited novelty and weak alignment with real-world scenarios. Compared with existing work on personalized multimodal retrieval, the novelty of the research problem is not clearly established, and there remains a gap to real-world application scenarios. The reviewers consider personalized multimodal retrieval to be a well-studied and widely adopted task, and the proposed setting does not provide sufficient differentiation. In addition, the personal photo album retrieval setup assumes extra textual input from users for learning, but no experiments are provided on datasets that reflect such user-generated annotations.

2. Insufficient methodological novelty and incomplete comparisons with prior work. The proposed approach mainly relies on query rewriting and RLHF techniques. However, reviewers noted that it lacks comparisons with strong existing query-rewriting methods, making the contribution appear more like an engineering integration rather than a substantial methodological advance.

3. Limited contribution of individual components and insufficient ablation analysis. The improvements brought by different internal modules are relatively small, and the paper lacks thorough ablation studies to validate their necessity. More detailed analysis is needed to justify the effectiveness of the proposed design.

**Reviewer Scores:**

The authors have responded to the reviewers’ concerns by adding new data scenarios. However, the task formulation still assumes that user-provided textual input is a fundamental source of information, which conflicts with reviewers’ concerns about the mismatch with real-world usage, where such input is often absent or unreliable. In addition, although additional experiments were provided, the reviewers’ requests for comparisons with query-rewriting baselines were not addressed; only qualitative analyses were added. As a result, I believe the likelihood that the reviewers will raise their scores is low.

---

### Decision · Program_Chairs · 2026-01-26

Reject